# CONSTRUCTIVE TT-REPRESENTATION OF THE TENSORS GIVEN AS INDEX INTERACTION FUNCTIONS WITH APPLICATIONS

**Gleb Ryzhakov,**
Skolkovo Institute of Science and Technology,
Moscow, Russia
`g.ryzhakov@skoltech.ru`

**Ivan Oseledets**
Skolkovo Institute of Science and Technology
and AIRI
Moscow, Russia
`i.oseledets@skoltech.ru`

## ABSTRACT

This paper presents a method to build explicit tensor-train (TT) representations. We show that a wide class of tensors can be explicitly represented with sparse TT-cores, obtaining, in many cases, optimal TT-ranks. Numerical experiments show that our method outperforms the existing ones in several practical applications, including game theory problems. Theoretical estimations of the number of operations show that in some problems, such as permanent calculation, our methods are close to the known optimal asymptotics, which are obtained by a completely different type of methods.

## 1 INTRODUCTION

The tensor train is a powerful tool for compressing multidimensional tensors (by tensor we mean a multidimensional array of complex numbers). It allows us to circumvent the curse of dimensionality in a number of cases. In a case of $d$-dimensional tensor with number of indices equal to $n$ for each dimension, direct storage of tensor involves $O(n^d)$ memory cells, while tensor train bypasses $O(ndr^2)$, where $r$ is average rank of TT decomposition (Oseledets, 2011). In many important applications, the average rank may be small enough so that $n^d \gg ndr^2$.

Tensor approximation is a hot topic in the area of machine learning. For example, in the paper (Richter et al., 2021) tensor train format is used to solve high-dimensional parabolic PDE with dimension in numerical experiments up to $d \sim 10^2$. Problems of building tensor decomposition and tensor completion are considered in (Lacroix et al., 2020; Fan, 2022; Ma & Solomonik, 2021). The properties of tensor decompositions as applied to machine learning tasks are discussed in (Ghalamkari & Sugiyama, 2021; Kileel et al., 2021; Khavari & Rabusseau, 2021).

Existing methods allow one to build TT-decompositions by treating the tensor values as a black box. The TT-cross approximation method (Oseledets & Tyrtyshnikov, 2010) adaptively queries the points where the tensor value is evaluated. The iterative alternative schemes such as alternating least squares method (Oseledets & Dolgov, 2012) or alternative linear schemes (Holtz et al., 2012), build a decomposition consistently updating the decomposition cores. These methods do not take into account the analytic dependence, if any, of the tensor value on its indices. At the same time, even for relatively simple tensors, these methods can build a TT decomposition for a long time and in the vast majority of cases obtain an answer with a given error greater than zero, even if the original tensor has an exact TT decomposition.

In this paper, we present a fast method to directly construct cores of the TT decomposition of a tensor for which the analytical[1] dependence of the tensor value on the values of its indices is known. Technically, our method works with functions, each of which depends on tensor index and which are sequentially applied to the values of the previous functions. This functions we call *derivative*

---

[1]By analytic dependence we mean the known symbolic formula for the tensor value, not the definition of the term within complex analysis.

*functions* hereafter. However, this assignment covers quite a large range of functional dependences of tensor value on its indices if such a set of functions is chosen skillfully. Examples are given in Section 3 and Appendix.

Our method works best in cases where the derivative functions together with the tensor itself have a small number of possible values. In the Application section and Appendix there are several examples for *indicator* tensors taking values only 0 and 1.

TT-cores, obtained by our method, are highly sparse, which gives an additional gain in performance. In many cases our method gives the lowest possible TT-rank, so that no further rounding of the TT-cores is required. In some other applications, the ranks of the TT decomposition obtained by our method can be substantially higher than those obtained by approximate methods. However, in a large number of such cases, the sparse structure of the cores allows one to achieve performance comparable to known algorithms.

The advantage of representing tensors in the TT format is not only in overcoming the curse of dimensionality, but also in the implemented tensor algebra for them: we can easily add, multiply, round TT-tensors, calculate the convolution (Oseledets, 2011). In this way we can, for example, construct a set of indicator tensors that represent some constraints in the given problem in advance, and then combine these constraints arbitrarily by multiplying these tensors with a data tensor. As a practical use of such a scheme, we give an example of calculating the permanent of a matrix. The cooperative games examples in Application section use the well-known algorithm for quickly finding the sum of all elements of the TT-tensor.

Other examples with practical problems are given in Appendix. They include: simple examples for sum, where we explicitly show sparse TT-cores; cooperative games, where we show how one can build iterative algorithm, based on our method; Knapsack problem (in several formulations), where we use existing algorithms to find the (quasi-) maximal element of the TT-tensor; Partition problem; Eight queens puzzle in several extended formulations (see Fig. 1 for the result for the case of 10 queens); sawtooth sequence; standard Boolean satisfiability problem.

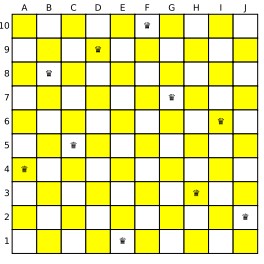

Figure 1: The solution of 10-queen problem, obtained by our algorithm.

Python code with the examples is available in the public domain[2]. In the vast majority of this examples we use products of tensors, convolution to find multidimensional sums and maximum element search to solve optimization problems. These operations are sufficient to solve a large class of problems from different areas of mathematics.

Our method has a direct extension to more complex cases of tensor networks, for one of the cooperative games below and in several examples in Appendix. Such a construction is called TT-Tucker (Dolgov & Khoromskij, 2013; Oseledets, 2011).

Our main contribution and advantages of our approach

- the exact and fast representation of the tensor in TT-format, which can then, if necessary, be rounded to smaller ranks with a given accuracy. In many of the given examples, this representation is optimal in the sense that the ranks of the TT decomposition cannot be reduced without loss of accuracy;

- highly sparse structure of TT-decomposition cores which leads to a noticeable reduction in calculations;

- a unified approach and a simple algorithmic interface to inherently different tasks and areas including those problems for which it is not immediately obvious the representation of the function specifying the tensor value in the form of consecutive functions (1)–(3);

- the ability to construct an approximate TT-decomposition with a controlled error or/and with the specified maximum ranks of the TT decomposition;

---

[2] https://github.com/G-Ryzhakov/Constructive-TT

- the possibility in some cases to explicitly reduce the set of matrix operations on the TT-cores in calculating the required value to an iterative algorithm, since the cores of TT-decomposition are sparse and their elements can be constructed explicitly in advance.

**Related works** In addition to the works mentioned above, let us list the following. In the paper (Oseledets, 2012) explicit representations of several tensors with known analytical dependence of indices are presented, bur for a fairly narrow class of tensors (see also (Khoromskij, 2018, Sec. 4.2)). In the survey (Grasedyck et al., 2013) techniques for low-rank tensor approximation are presented, including TT-format. In works (Cichocki et al., 2016; 2017) many examples of applying Tucker and Tensor Train decomposition to various problems including machine learning and data mining algorithms. Tensor completion method described in the paper (Steinlechner, 2016) uses Riemannian optimization for reconstruction a TT-format of a blackbox. In the paper (Bian et al., 2022) energy-based learning approach for cooperative games is considered.

**Background** Consider a tensor $\mathbf{K}$ with the following dimensions $\{n_1, n_2, \ldots, n_d\}$, i. e., $\mathbf{K} \in \mathbb{C}^{n_1 \times n_2 \times \cdots \times n_d}$, where $\mathbb{C}$ is the set of complex numbers. TT-decomposition of the tensor $\mathbf{K}$ with *TT-ranks* $\{r_0, r_1, \ldots, r_d\}$ is defined as the product

$$K(i_1, i_2, \ldots, i_n) = \sum_{\alpha_0=1}^{1} \sum_{\alpha_1=1}^{r_1} \cdots \sum_{\alpha_{d-1}=1}^{r_{d-1}} \sum_{\alpha_d=1}^{1} G_1(\alpha_0, i_1, \alpha_1) G_2(\alpha_1, i_2, \alpha_2) \cdots$$
$$\cdots G_{d-1}(\alpha_{d-2}, i_{d-1}, \alpha_{d-1}) G_d(\alpha_{d-1}, i_d, \alpha_d),$$

where tensors $\mathbf{G}_i \in \mathbb{C}^{r_{i-1} \times n_i \times r_i}$ are called *cores* of the TT-decomposition (we let $r_0 = r_d = 1$).

## 2 BUILDING TT-REPRESENTATION WITH THE GIVEN SEQUENCE OF FUNCTIONS

Note that if one needs to calculate the value of the TT-tensor in a given multi-index $\{i_1, \ldots, i_d\}$, it is more efficient to start the calculation from one end, as described in (Oseledets, 2011). In this approach, we take the vector $\boldsymbol{v}_1 := \mathbf{G}_1(1, i_1, :)$ and multiply it by the matrix $\mathbf{G}_2(:, i_2, :)$. Then we multiply the obtained vector $\boldsymbol{v}_2$ by the matrix $\mathbf{G}_3(:, i_3, :)$, etc. At each step we get $\boldsymbol{v}_k = \boldsymbol{v}_{k-1} \mathbf{G}_k(:, i_k, :)$. The above process can be viewed as a process of sequential transformation (in this case, linear) of the vector obtained at the previous step. Our algorithm is based on the idea of the inverse step: knowing the sequence of arbitrary (in particular, nonlinear) transformations, can we obtain cores of TT-decomposition, each of which corresponds to its sequential transformation?

In addition, there is the problem of reducing an arbitrary function of many variables to a sequence of functions, each of which depends on only one index and the value of the previous function.

### 2.1 DERIVATIVE FUNCTIONS

From an algebraic point of view, we want to build TT-decomposition of such tensors $\mathbf{K}$, each element $K(i_1, i_2, \ldots, i_d)$ of which can be calculated in two consecutive passages. One, from to the left of the right, is as follows

$$a_1(i_1) = f^{(1)}(i_1, 0), \quad a_2(i_1, i_2) = f^{(2)}(i_2, a_1), \quad a_3(i_1, i_2, i_3) = f^{(3)}(i_3, a_2),$$
$$\cdots$$
$$a_{l-1}(i_1, i_2, \cdots, i_{l-1}) = f^{(l-1)}(i_{l-1}, a_{l-2}), \tag{1}$$

then from right to left

$$a_d(i_d) = f^{(d)}(i_d, 0), \quad a_{d-1}(i_d, i_{d-1}) = f^{(d-1)}(i_{d-1}, a_d),$$
$$\cdots$$
$$a_{l+1}(i_d, i_{d-1}, \cdots, i_{l+1}) = f^{(l+1)}(i_{l+1}, a_{l+2}), \tag{2}$$

and, finally,

$$K(i_1, i_2, \ldots, i_d) = f^{(l)}(i_l, a_{l-1}, a_{l+1}). \tag{3}$$

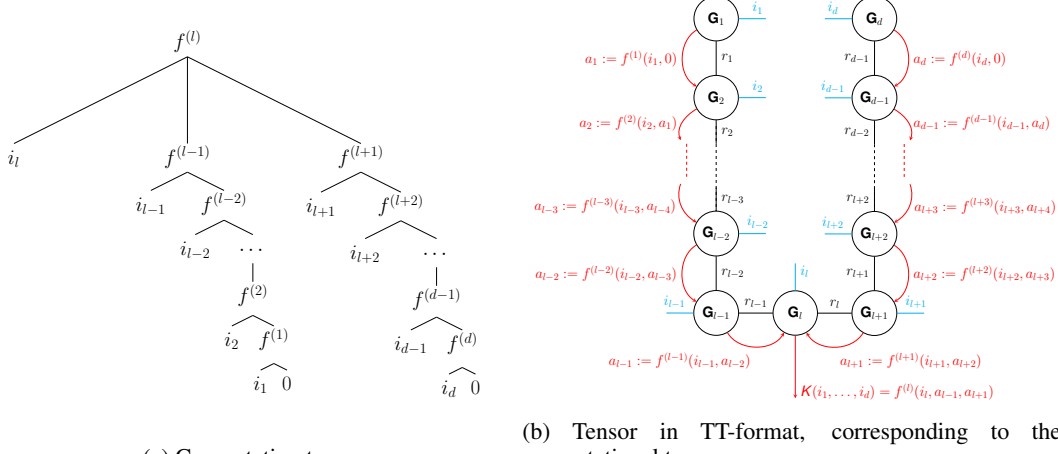

(a) Computation tree

(b) Tensor in TT-format, corresponding to the computational tree

Figure 2: Computation tree we can handle and the resulting TT-decomposition

The computation tree of this procedure is shown on Fig. 2a.

The underlying idea of reducing a non-linear transformation $f^{(k)}(i_k, a_{k-1}) \to a_k$ to a linear one $\boldsymbol{v}_{k-1} \mathbf{G}_k(:, i_k, :) \to \boldsymbol{v}_k$ is quite simple: we assign to each possible input and output value of the derived function a different basic vector $\boldsymbol{e}$, then the specified transformation can be represented as matrix-vector multiplication (see Theorem 2.1 below). Thus, each function $f^{(k)}$ corresponds to a core in the TT-decomposition of the resulting tensor, see Fig. 2b. In this figure, the cores of the expansion, which are 3-dimensional tensors, are represented by circles. The blue lines with indices correspond to the input indices of the tensor. The black lines connecting the cores correspond to the dimensions, by which the summation takes place. The red arrows show the correspondence between the output vector obtained by successive multiplication of the cores, starting from the left or right end, and the values of the derivative functions.

We write the first argument of the derivative functions $f^{(k)}$ as a lower index when describing examples and applications.

## 2.2 REPRESENTATION OF A MULTIVARIATE FUNCTION AS A SEQUENCE OF FUNCTIONS

Unfortunately, there is no single algorithmic approach for constructing a sequence of derivative functions based on a given analytical dependence of the tensor value on its indices. Moreover, such a representation is not unique and different representations can lead to different ranks of the resulting tensor. Several examples of different sets of derivative functions for the same tensor are given further in the section with experiments and in Appendix. However, it turns out that there are fairly general patterns for derivative functions, and we hope that, based on the examples we give from various applications of the method, it is easy to construct by analogy the desired sequence of derivative functions for a particular application problem.

As an example, consider step function in the so-called *Quantized Tensor Train decomposition* (QTT) ((Oseledets, 2010; Khoromskij, 2011)) when the tensor indices are binary $i_k \in \{0, 1\}$ and all together represent a binary representation of some integer from the set $\{0, 1, \ldots, 2^d - 1\}$. The value of the tensor $\boldsymbol{I}$ represents the value of some given function $P$ defined on this set, with function $P$ equal to the step function $P = P_{\text{step}}$ in this example:

$$I(i_1, i_2, \ldots, i_d) = P\left(\sum_{j=0}^{d-1} i_{d-i} 2^j\right), \quad P_{\text{step}}(x) = \mathbf{1}_{x>t} := \begin{cases} 0, & x \le t, \\ 1, & x > t \end{cases} \tag{4}$$

for the given integer number $t, 0 \le t < 2^d$. Let the binary representation of $t$ be $t = \sum_{j=0}^{d-1} b_{d-i} 2^j$. Then the form of the derivative functions for this tensor are depend only on the value of $b_k$ and do not depend on the index $k$ itself. This function are the following:

If $b_k = 0$, then $f_0^k(x) := f^{(k)}(0, x) = x,$ $\qquad\qquad f_1^k(x) := f^{(k)}(1, x) = 1;$

$$\text{if } b_k = 1, \quad \text{then } f_0^k(x) := f^{(k)}(0,\, x) = \begin{cases} 1, & x = 1 \\ \text{None}, & x = 0. \end{cases}, \qquad f_1^k(x) := f^{(k)}(1,\, x) = x.$$

In our method, the functions $f$ are predefined in the following way. If, in the process of calculating a tensor value, the function $f^{(k)}$ arguments are not in its domain, we assume that it returns an empty value (we denote this value by None as in Python language). The next function ($f^{(k-1)}$ or $f^{(k+1)}$), having received None, returns also None, and so on, up to the "middle" function $f^{(l)}$, which returns 0 if at least one of its arguments is None. A detailed explanation with technical details of how such a set of derivation functions leads to the step function is given in the Appendix.

In this example, the "middle" function is the last function: $l = d$, thus we consider it as a function of two arguments. The same is true for other examples in which the "middle" function is the first or last.

Note that in this example, the original analytic representation for the tensor did not assume pairwise interaction of the indices. On the contrary, the formula (4) is quite integral: its value depends on all variables at once. Nevertheless, the expressions for the derivative function turned out to be simple enough. This situation holds for many examples as well, see Applications and Appendix. Thus, our method can find a wide application.

It is worth noting that the arguments of the functions $f$ have different physical meaning. The first argument is the original natural number, it corresponds to an index of the tensor under consideration. By contrast, the second argument and the function value itself determine the relation between two indices of the tensor and this relation can be complex. In the general case, middle-function $f^{(l)}$ is complex-valued, and the values of all other function $f^{(k)}$, $k \neq l$, can be of any nature for which a comparison operation is defined.

## 2.3 TT-DECOMPOSITION WITH THE GIVEN DERIVATIVE FUNCTIONS

**Theorem 2.1.** *Let $\mathbb{D}_j$ be the image of the $j$-th function $f^{(j)}$ from the derivative function set for $1 \leq j < l$ ("left" function):*

$$\mathbb{D}_j = \left\{ f^{(j)}(i,\, x) \colon 1 \leq i \leq n_i,\, x \in \mathbb{D}_{j-1},\ \text{if } f^{(j)} \text{ is defined at } (i,\, x) \right\}, \quad j = 1,\, \ldots,\, l-1.$$

*We let $\mathbb{D}_0 = \{0\}$. Similarly for the "right" functions with $l < j \leq d$:*

$$\mathbb{D}_j = \left\{ f^{(j+1)}(i,\, x) \colon 1 \leq i \leq n_i,\, x \in \mathbb{D}_{j+1},\ \text{if } f^{(j+1)} \text{ is defined at } (i,\, x) \right\}, \quad j = l,\, \ldots,\, d-1,$$

*where we let $\mathbb{D}_d = \{0\}$. Then there exists TT-representation of the tensor $\mathbf{K}$ (3) with TT-ranks $r$ not greater than*

$$r = \{ |\mathbb{D}_0| = 1,\, |\mathbb{D}_1|,\, \ldots,\, |\mathbb{D}_{d-1}|,\, |\mathbb{D}_d| = 1 \},$$

*where $|\mathbb{A}|$ denotes the cardinality of a set $\mathbb{A}$.*

*Proof.* The proof is constructive. The construction of the required cores of the TT-decomposition takes place in two stages.

We first enumerate elements of all images $\{\mathbb{D}_i\}_{i=0}^d$ in arbitrary order, such that we can address them by index. Denote $\mathbb{D}_j[n]$ the $n$-th element of $j$-th image, $n = 1,\, \ldots,\, |\mathbb{D}_j|$. Now we can switch from the initial functions $f$ with arbitrary values to functions $\hat{f}$, the range of each is a consecutive set of natural numbers starting from 1:

$$\hat{f}^{(j)}(i,\, x) := \text{index\_of}\left( f^{(j)}\left(i,\, \mathbb{D}_{j-1}[x]\right),\, \mathbb{D}_j \right), \quad j = 1,\, \ldots,\, l-1,\ x = 1,\, \ldots,\, |\mathbb{D}_{j-1}|$$

$$\hat{f}^{(j)}(i,\, x) := \text{index\_of}\left( f^{(j)}\left(i,\, \mathbb{D}_{j+1}[x]\right),\, \mathbb{D}_j \right), \quad j = l+1,\, \ldots,\, d,\ x = 1,\, \ldots,\, |\mathbb{D}_{j+1}|, \quad (5)$$

where function index\_of is defined as follows

$$z = \text{index\_of}(y,\, \mathbb{A}) \iff y = \mathbb{A}[z] \quad \text{for some ordered set } \mathbb{A} \text{ and any } y \in \mathbb{A}.$$

We let index\_of(None, $\mathbb{A}$) := None. Function $\hat{f}^{(l)}$, which corresponds to the "middle" function $f^{(l)}$, is defined as follows $\hat{f}^{(l)}(i,\, x,\, y) := f^{(l)}\left(i,\, \mathbb{D}_{l-1}[x],\, \mathbb{D}_l[y]\right)$.

In the second stage, we assign each integer input and output of new functions $\{\hat{f}_i\}$ to a corresponding basis vector $e$[3]. The basic idea is to construct the $j$-th "left" core ($j < l$) of the desired TT-decomposition corresponding to the function $\hat{f}^{(j)}$ according to the following scheme:

$$\text{if } y = \hat{f}^{(j)}(i, x), \quad \text{then } (e^{(x)})^T \mathbf{G}_j(:, i, :) = (e^{(y)})^T, \quad i = 1, \ldots, n_j \quad (6)$$

where $\mathbf{G}_j(:, i, :) \in \mathbb{R}^{|\mathbb{D}_{j-1}| \times |\mathbb{D}_j|}$ denotes the matrix representing the $i$-th slice of $j$-th core. The elements of this core are constructed explicitly:

$$G_j(x, i, y) = \begin{cases} 1, & \text{if } y = \hat{f}^{(j)}(i, x) \\ 0, & \text{else} \end{cases}, \quad x = 1, \ldots, |\mathbb{D}_{j-1}|, \ y = 1, \ldots, |\mathbb{D}_j|$$

We do the same for the "right" cores or which $i > l$, except that multiplication on the basis vector takes place on the right: if $y = \hat{f}^{(j)}(i, x)$, then $\mathbf{G}_j(:, i, :)e^{(x)} = e^{(y)}, i = 1, \ldots, n_j$.

Finally, we construct the middle-core $\mathbf{G}_l$ which corresponds to the function $f^{(l)}$:

$$\mathbf{G}_l(:, i, :) = \begin{cases} \hat{f}^{(l)}(i, x, y), & \text{if } \hat{f}^{(l)} \text{ defined on } (i, x, y) \\ 0, & \text{else} \end{cases}, \quad x = 1, \ldots, |\mathbb{D}_{l-1}|, \ y = 1, \ldots, |\mathbb{D}_l|.$$

We summarize this two stages in Algorithms 1–2 in Appendix.

Theorem statement follows from this construction: after multiplying $m$ "left" constructed cores, $1 \le m < l$, starting from the first one, we get the following basis vector: $\mathbf{G}_1(1, i_1, :)\mathbf{G}_2(:, i_2, :) \cdots \mathbf{G}_m(:, i_m, :) = (e^{(a_m)})^T$, where $a_m$ is defined in (1). A similar basis vector $e^{(a_p)}$ is obtained by successive multiplication of all cores, starting from the last one with the index $d$ and up to some $p > l$ with $a_p$ defined in (2). Conclusively, the statement of the theorem follows from the relation $e^{(a_{l-1})}\mathbf{G}_l(:, i_l, :)e^{(a_{l+1})} = K(i_1, i_2, \ldots, i_d)$ which is a consequence of the definition of the elements of the middle-core $\mathbf{G}_l$ and which corresponds to the relation (3). $\qquad\square$

**Rank reduction** One way to reduce the TT-rank in the case when an image $\mathbb{D}_i$ of a function $f^{(i)}$ contains too many elements is to partition the image $\mathbb{D}_i$ into several sets and map the basis vector $e$, discussed in the second stage of the Theorem, to one of these sets. This is possible if the value of the function belongs to a space with a given topology. In the simplest case, when the value of the function is real, we can combine into one specified set only those elements from the image of the function for which $|x - y| < \epsilon, x, y \in \mathbb{D}_i$ is satisfied with the given $\epsilon$. In addition, we can specify a maximum number of such sets, thus fixing the maximum rank, increasing the value of $\epsilon$ or combining some sets with each other. Other ways of reducing the rank are described in the Appendix.

**Complexity** The cores, except perhaps the middle-core, obtained by our method are highly sparse. Each row of the slice $\mathbf{G}(:, i, :)$ of the core to the left of the middle-core consists of zeros, except maybe one unit. The same is true for the columns of the core to the right of the middle-core. When multiplying a slice of a core by a vector, we consider only those operations which do not involve addition with zero or multiplication by zero or unit.

When using the compressed format, formally we do not need addition and multiplication to obtain a single tensor element, since its calculation is reduced only to choosing the desired slice indices.

Consider a TT-tensor $\mathbf{G}$ obtained by our method with dimensions $\{n_1, n_2, \ldots, n_d\}$ and ranks $\{r_0, r_1, \ldots, r_d\}$ with the middle-core at the position $l$, $1 \le l \le d$. To calculate the convolution of the tensor $\mathbf{G}$ and an arbitrary rank-one tensor $\mathbf{W}$ of the form

$$\langle \mathbf{G}, \mathbf{W} \rangle = \sum_{r_1, r_2, \ldots, r_{d-1}} \sum_{n_1, n_2, \ldots, n_d} G_1(1, n_1, r_1) \ldots G_d(r_{d-1}, n_d, 1) w(n_1) \cdots w(n_d), \quad (7)$$

where $w(n) := W(1, n, 1)$, we need no more than $n_{\text{conv}}$ additions and no more than $n_{\text{conv}}$ multiplications with $n_{\text{conv}} = \sum_{i=1}^{l-1} n_i r_{i-1} + \sum_{i=l+1}^{d} n_i r_i + n_m \big( r_{l-1} r_l + \min(r_{l-1}, r_l) \big)$. Indeed, the first two sums in the last expression correspond to successive multiplication of the vector by the current core slice for successive multiplication from each end of the tensor train up to the middle kernel with index $l$. The last term corresponds to the multiplication of the two resulting vectors by the middle-core (which we assume dense) left and right.

---

[3]column of the identity matrix of appropriate size.

## 3 APPLICATIONS

As a practical application of our method, in this section we give: a) examples from the field of cooperative games, where we compare with existing methods and show our superiority both in execution time and in accuracy (our algorithm gives machine precision); b) the problem of calculating matrix permanent, where our method gives an estimated number of operations only twice as large as the optimized *ad hoc* method of calculating the permanent using Hamiltonian walks.

### 3.1 COOPERATIVE GAMES EXAMPLES

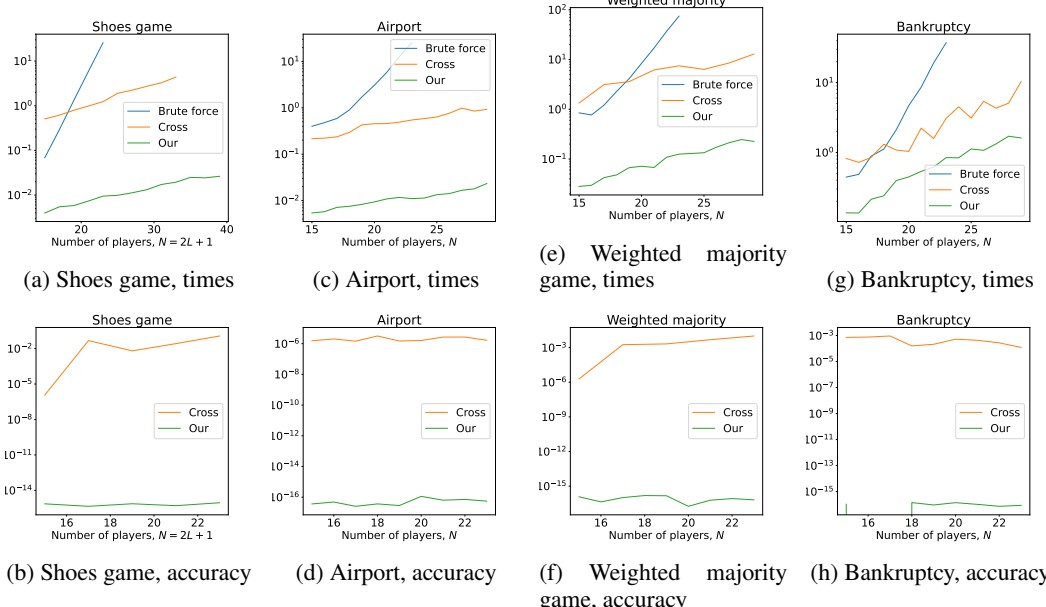

Figure 3: Times in seconds and relative accuracy as functions of number of players for four cooperative games. Brute force—calculating the sum (8) directly, Cross—results from the paper (Ballester-Ripoll, 2022).

As an example, consider several so-called cooperative games (von Neumann & Morgenstern, 2007). Omitting the details of the economic formulation of the problem, let us briefly consider its mathematical model. In general, in the theory of cooperative games it is necessary to calculate the following sum over all subsets of the given set $\mathbb{T}$ of players

$$\pi(k) := \sum_{\mathbb{S} \subseteq \mathbb{T} \setminus \{k\}} p(|\mathbb{S}|)\big(\nu(\mathbb{S} \cup \{k\}) - \nu(\mathbb{S})\big), \quad \text{for all } k \in \mathbb{T}. \tag{8}$$

Here $p$ is some function of the number of players in a *coalition* $\mathbb{S}$. The function of a coalition $\nu$ is the function of interest, it depends on the game under consideration. This function denotes the gain that a given coalition receives (*value of the coalition*).

Below we briefly review several cooperative games and compare the performance and accuracy of our algorithm for solving them with the algorithm presented in (Ballester-Ripoll, 2022), which is based on the TT-cross approximation method [4]. Due to space constraints, we give only a brief statement of the problem, the constructed derivative functions and the results of our method. A detailed description of the application of our approach to cooperative games can be found in Appendix. The results of the algorithm comparison are shown in Fig. 3. In all the examples shown in this figure our algorithm produces results with machine accuracy and faster than the one from the cited paper.

---

[4]We use the same setup as described in the cited paper for the experiments, and take code from the open source repository `https://github.com/rballester/ttgames/` of the author of this paper.

**Shoe sale game.** In this game, participants are divided into two categories—those who sell left boots (indices 1 through $L$) and those who sell right boots (indices $L + 1$ through $2L + 1$). As shoes can be sold only in pairs, the value of a coalition is the minimum of the numbers of "left" and "right" players in a coalition.

Let us build tensors for this game. To find the required value $\pi$ (8) in the case of cooperative games, it is convenient to construct tensors that have a dimension equal to the number of players. Each index of this tensor is binary: 1 means a player is a member of a coalition, 0 means he is not.

To construct the TT decomposition of the tensors $p(|\mathbb{S}|)\nu(\mathbb{S})$ using our method, let us take the following derivative functions:

$$f_i^{(k)}(x) = x + i, \ \ 1 \le k \le d, \ k \ne L+1, \quad f_i^{(L+1)}(x, y+i) = \min(x, y)p(x+y+i), \ \ i = 0, 1,$$

thus middle-core is placed on the position $l = L + 1$. The derivative functions for constructing the tensor $p(|\mathbb{S}| - 1)\nu(\mathbb{S})$ are selected in a similar way (we let $p(-1) = p(2L + 1) = 0$). Once the cores for both tensors are constructed, we can calculate the sum (8) for different values of $k$ by taking the corresponding slices of the cores by the method described in (Ballester-Ripoll, 2022) and performing the convolution.

**Airport.** It is not a cooperative game in its purest form, as instead of gain we have a payoff, but the mathematical essence is in the spirit of cooperative games. Each player represents an aircraft that needs a landing strip of length $c_k$. Thus, $\nu(\mathbb{S}) = \max\{c_i : i \in \mathbb{S}\}$. In order to construct a TT-representation of the tensor corresponding to this $\nu$, let us first order the values of $c_k$ in descending order. Then the derivative functions are

$$f_0^k(x) = x, \quad f_1^k(x) = \begin{cases} x, & x > 0 \\ c_k, & x = 0, \end{cases}, \quad 1 \le k \le d.$$

However, with these derivative functions the TT-ranks can get very large, especially for non-integer $c_k$. To reduce the ranks, we do the following trick: we break the second function $f_1^k$ into two terms,

$$f_1^k(x) = f_{(1)}^k(x) + c_k \cdot f_{(2)}^k(x), \quad f_{(1)}^k(x) = \begin{cases} x, & x > 0 \\ \text{None}, & \text{else} \end{cases}, \quad f_{(2)}^k(x) = \begin{cases} 1, & x = 0 \\ \text{None}, & \text{else} \end{cases}.$$

After that, we use three derivative functions to build TT-cores, taking the multipliers $c_k$ out of the build core as shown in Figure 4a. This gives us a TT-Tucker format with TT-ranks equal to 2 and matrices $\boldsymbol{A}_k$ with the following elements: $\boldsymbol{A}_k = \begin{pmatrix} 1 & 0 & 0 \\ 0 & 1 & c_k \end{pmatrix}$.

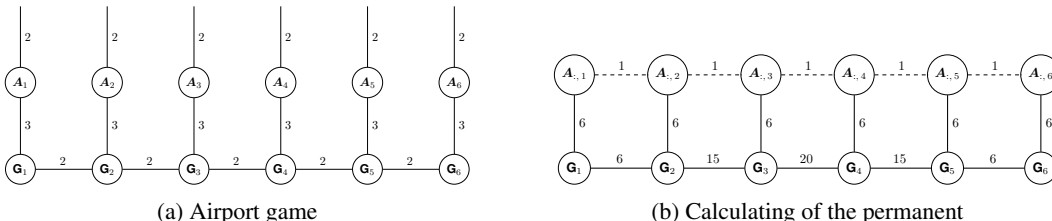

|  (a) Airport game  |  (b) Calculating of the permanent  |

Figure 4: Tensor network for two problems, $d = 6$. The numbers denote ranks (number of terms in sum). The dashed line shows that at this point the rank is 1, which means that there is no summation and the corresponding cores are independent.

**Other games.** Fig. 3 also shows comparisons with the other two games. In Weighted majority game tensor $\nu$ has the form $\nu(\mathbb{S}) = 1$ if $\sum_{i \in \mathbb{S}} w_i \ge M$ and 0 otherwise for given weights $\{w_i\}$ and threshold $M$. In Bankruptcy game, $\nu(\mathbb{S}) = \max(0, E - \sum_{i \notin \mathbb{S}} c_i)$ for the given values of $\{c_i\}$ and $E$. The derivative functions for these problems are chosen quite straightforwardly.

### 3.2 MATRIX CALCULUS: PERMANENT

Consider a task of calculating permanent of a matrix $\{a_{ij}\} = A \in \mathbb{C}^{d \times d}$. To solve this problem using the presented technique, let us construct two tensors in TT-format. The first tensor $\mathbf{A}$ will

represent products of matrix $A$ elements in the form: $A(i_1, i_2, \ldots, i_d) = A_{i_1 1} A_{i_2 2} \cdots A_{i_d d}$. This is rank-1 tensor and its cores $\{H_k \in \mathbb{C}^{1 \times d \times 1}\}_{k=1}^{d}$ are $H_k(1, i, 1) = A_{ik}, i = 1, \ldots, d$.

The second tensor $I$ is an indicator tensor for such a set of indices, in which all indices are different

$$I(i_1, i_2, \ldots, i_d) = \begin{cases} 1, & \text{if all } i_1, i_2, \ldots, i_d \text{ are different,} \\ 0, & \text{else.} \end{cases}$$

The cores $G$ of this tensor are obtained using the following derivative functions

$$f_i^k(x) = \begin{cases} x + 2^i, & x \,\&\, 2^i = 0, \\ \texttt{None}, & \text{else} \end{cases}, \quad k < d; \quad f_i^d(x) = \begin{cases} 1, & x \,\&\, 2^i = 0, \\ 0, & \text{else} \end{cases},$$

where the ampersand sign stands for bitwise AND for integers. In this scheme the middle-core is the last ($d$-th) core. The $k$-th bit of the input $x$ of the derivative functions corresponds to the $i$-th value of the set $\{1, 2, \ldots, d\}$: if this bit is zero, then the $i$-th value of the index has not occurred before, and thus this value is valid for the current index. The function sets this bit in $x$ and the value $x$ is passed on. If the bit has already been set, the derivative function returns $\texttt{None}$, since two identical indexes are forbidden.

The permanent value is equal to the convolution of tensor $I$ and tensor $A$. Since the tensor $A$ is one-rank tensor, we can look at the computation of the permanent as a contraction of the tensor $I$ with weights equal to the corresponding elements of the given matrix $A$, see Fig. 4b.

After calculating the function $f_i^k$ we get a number $x$ in the binary representation of which there are exactly $k$ bits equal to one. From this one can concludes that the corresponding rank $r_k$ is equal to $r_k = \frac{d!}{(d-k)!k!}$. Using relation (7) we can obtain an upper estimate on the total number of operations for the convolution of the given tensor as $2\overline{n}_{\text{conv}} \sim (2^{N+1}N)$. However, one can notice that on average half of the rows in the cores slices consist entirely of zeros, since index repetition is "forbidden": $G_k(i, j, :) = 0$ at index $i$ corresponds to the $x$ with the $j$-th bit set.

Thus, $n_{\text{conv}} = 1/2\overline{n}_{\text{conv}}$ and after the cores are built, total number of operations $n_{\text{tot}}$ (both additions and multiplications) required to obtain the result at these ranks has asymotics $n_{\text{tot}} = 2n_{\text{conv}} \sim 2^N N$. This asymptotic is better than the one that can be obtained from the well-known Ryser's formula for calculating the permanent: $P(A) = (-1)^N \sum_{\mathbb{S} \subseteq \{1,2,\ldots,N\}} (-1)^{|\mathbb{S}|} \prod_{i=1}^{N} \sum_{j \in \mathbb{S}} A_{ij}$. When applied head-on, this formula requires $O(2^{N-1}N^2)$ operations. It is true that if one uses Hamiltonian walk on $(N-1)$-cube (Gray code) for a more optimal sequence of subset walks, this formula will give asymptotic $n_{\text{tot}} \sim (2^{N-1}N)$ which is only twice as good as ours (Nijenhuis & Wilf, 1978, pp. 220–224).

This is an example of a problem where we can first pre-calculate the tensor $I$ with conditions and then reuse it with different data tensors $A$ containing elements of a particular matrix.

## 4 CONCLUSIONS AND FUTURE WORK

We presented an algorithm for constructing the tensor in TT-format in the case when an explicit analytic dependence is given between the indices. The cores of the obtained TT-tensor are sparse, which speeds up manipulations with such a tensor. Examples are given in which our method can be used to construct TT-representations of tensors encountered in many applied and theoretical problems. In some problems, our representation yields an answer faster and more accurately than state-of-the-art algorithms. Thus, in the vast majority of cases, we are not faced with a trade-off between speed and accuracy.

As a limitation of our method, let us point out the fast growth of the ranks in the case when the derivative functions have a large size of their image set. Although we have a rank restriction procedure for this case, as plans for the future we specify an extension of our algorithm to accurately construct a TT-decomposition for such cases as well, if it is known to be low-ranked.

### ACKNOWLEDGEMENTS

The work was supported by the Analytical center under the RF Government (subsidy agreement 000000D730321P5Q0002, Grant No. 70-2021-00145 02.11.2021).

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

APPENDICES

## A ALGORITHMS

Algorithms 1–2 summarize the constructive construction of TT-decomposition cores, which is described in Theorem 2.1. Here, the function `order` orders the set in any way, `dom` denotes the domain of a function. If a function returns `None` on some set of its arguments, then we assume that it is not defined on that set.

## B OTHER APPLICATIONS

The examples in this section fall into two (possibly intersecting) broad types. The first of them contains the calculation of the exact value of some quantity for which the TT decomposition of the tensor arising in the problem is needed. In this case, we do not perform a TT tensor rounding (except when we reduce the rank using SVD decomposition with zero threshold, but we assume that after that the tensor value remains the same within machine accuracy). From a technical point of view, usually the convolution operations (or other similar operations) are performed without explicitly constructing the cores of TT-decomposition completely in computer memory. Instead, we use the functions obtained after applying the Algorithm 1 directly as arrays of their values. This is equivalent to working with matrices in a compressed format.

The second type of application consists of problems in which rounding with a given accuracy is the advantage of the tensor approach—we get the answer with a certain error, but faster.

### B.1 SIMPLE (MODEL) EXAMPLES

**Sum** Consider the following tensor $\mathbf{S}$, which is some function $P$ of the sum of the elements of the given vectors $\boldsymbol{a}_1$, $\boldsymbol{a}_2$, ..., $\boldsymbol{a}_d$:

$$I(i_1, i_2, \ldots, i_d) = P(\boldsymbol{a}_1[i_1] + \boldsymbol{a}_2[i_2] + \ldots + \boldsymbol{a}_d[i_d]).$$

We can easily build its TT-representation using the presented technique if we put the derivative functions equal to

$$f_i^k(x) = x + \boldsymbol{a}_k[i].$$

The view of these functions is the same for all cores except for the middle-core. The middle-core can stand in any place in this case, but it makes sense to put it in the middle of the tensor train (at the position $\lfloor (d+1)/2 \rfloor$) to reduce the TT-ranks. For the middle-core on the $m$-th place the derivative functions are equal to

$$f_i^m(x, y) = P(x + y + \boldsymbol{a}_m[i]).$$

In the simple case, when the vector elements are consecutive numbers from zero to a given number: $\boldsymbol{a}_i = \{0, 1, \ldots, b_i\}$, we have a natural correspondence between the value of the function $y$ and the basis vector $e^T(i)$ representing it:

$$y \Longleftrightarrow \boldsymbol{e}^{(y+1)}$$

Thus, for example, the third slice (of any of first $m-1$ cores) corresponding to the addition of number 2 will be of the form

$$\mathbf{G}(:, 3, :) = \begin{pmatrix} 0 & 0 & 0 & 0 & 0 & \cdots & 0 & \cdots & 0 \\ 0 & 0 & 0 & 0 & 0 & \cdots & 0 & \cdots & 0 \\ 1 & 0 & 0 & 0 & 0 & \cdots & 0 & \cdots & 0 \\ 0 & 1 & 0 & 0 & 0 & \cdots & 0 & \cdots & 0 \\ 0 & 0 & 1 & 0 & 0 & \cdots & 0 & \cdots & 0 \\ & & & \vdots & & & & & \\ 0 & 0 & 0 & 0 & 0 & \cdots & 1 & \cdots & 0 \end{pmatrix},$$

---

**Algorithm 1** Construction of the integer-valued functions based on the given complex-valued functions

---

**Require:** Middle-index $l$, set of functions $\{f^{(i)}\}_{i=1}^{d}$ of two variables (function $f^{(l)}$ have 3 arguments)

**Ensure:** Functions $\{\hat{f}_j^{(i)}\}$

1: # *Initialization*
2: $n_k$ = maximum of the domain of the function $f^{(k)}$ on first variable for $k = 1, \ldots, d$ (dimensions of the resulting tensor)
3: # *Part I. Finding the function outputs $R$ of each function. $R$ is a list of arrays.*
4: $R[0] \leftarrow \{0\}$, $R[d] \leftarrow \{0\}$
5: **for** $i = 1$ to $l - 1$ **do**
6:     $R[i] \leftarrow \text{order}(\{f^{(i)}(k, x) \colon k = 1, \ldots, n_i, \, x \in R[i-1]\})$
7: **end for**
8: **for** $i = d - 1$ to $l$ step $-1$ **do**
9:     $R[i] \leftarrow \text{order}(\{f^{(i)}(k, x) \colon k = 1, \ldots, n_i, \, x \in R[i+1]\})$
10: **end for**
11: # *Part II. Defining new functions*
12: # *From the left*
13: **for** $i = 1$ to $l - 1$ **do**
14:     **for** $j = 1$ to $n_i$ **do**
15:         **for** $k$ in $R[i-1]$ **do**
16:             **if** defined $f_j^{(i)}(k)$ **then**
17:                 $x \leftarrow \text{index\_of}(k, \, R[i-1])$
18:                 $y \leftarrow \text{index\_of}(f^{(i)}(j, k), \, R[i])$
19:                 $\hat{f}_j^{(i)}(x) := y$
20:             **end if**
21:         **end for**
22:     **end for**
23: **end for**
24: # *From the right*
25: **for** $i = d$ to $l + 1$ step $-1$ **do**
26:     **for** $j = 1$ to $n_i$ **do**
27:         **for** $k$ in $R[i]$ **do**
28:             **if** defined $f_j^{(i)}(k)$ **then**
29:                 $x \leftarrow \text{index\_of}(k, \, R[i])$
30:                 $y \leftarrow \text{index\_of}(f^{(i)}(j, k), \, R[i-1])$
31:                 $\hat{f}_j^{(i)}(x) := y$
32:             **end if**
33:         **end for**
34:     **end for**
35: **end for**
36: # *Middle-index function*
37: **for** $j = 1$ to $n_l$ **do**
38:     **for** $k_1$ in $R[l-1]$, $k_2$ in $R[l]$ **do**
39:         **if** defined $f^{(l)}(j, k_1, k_2)$ **then**
40:             $x_1 \leftarrow \text{index\_of}(k_1, \, R[l-1])$
41:             $x_2 \leftarrow \text{index\_of}(k_2, \, R[l])$
42:             $\hat{f}_j^{(l)}(x_1, x_2) := f^{(l)}(j, k_1, k_2)$
43:         **end if**
44:     **end for**
45: **end for**
46: Return functions $\{\hat{f}_j^{(i)}\}$, (optionally) outputs $R$.

---

---

**Algorithm 2** Explicit construction of the cores of the functional tensor

---

**Require:** Middle-index $l$, set of integer-valued functions $\{\hat{f}_j^{(i)}\}$ of one variable (function $\{\hat{f}_j^{(l)}\}$ have 2 arguments)
**Ensure:** Cores $\mathbf{G}_1, \ldots, \mathbf{G}_d$ of TT-decomposition of the functional tensor
 1: # *Initialization*
 2: $n_k =$ maximum of the index $j$ value of the function $\hat{f}_j^{(k)}$ for $k = 1, \ldots, d$
 3: # *From head*
 4: **for** $i = 1$ to $l - 1$ **do**
 5:     $\mathbf{G}_i \leftarrow \text{zeros}\big(\max_j \sup(\text{dom }\hat{f}_j^{(i)}), n_i, \max_{j,\,x} \hat{f}_j^{(i)}(x)\big)$
 6:     **for** $j = 1$ to $n_i$ **do** #*Build each slice of the core as in* (6)
 7:         **for** $x$ in dom $\hat{f}_j^{(i)}$ **do**
 8:             $G_i(x,\, j,\, \hat{f}_j^{(i)}(x)) \leftarrow 1$
 9:         **end for**
10:     **end for**
11: **end for**
12: # *From tail*
13: **for** $i = d$ to $l + 1$ step $-1$ **do**
14:     $\mathbf{G}_i \leftarrow \text{zeros}\big(\max_{j,\,x} \hat{f}_j^{(i)}(x), n_i, \max_j \sup(\text{dom }\hat{f}_j^{(i)})\big)$
15:     **for** $j = 1$ to $n_i$ **do**
16:         **for** $x$ in dom $f_j^{(i)}$ **do**
17:             $G_i(\hat{f}_j^{(i)}(x),\, j,\, x) \leftarrow 1$
18:         **end for**
19:     **end for**
20: **end for**
21: # *Build middle-index core*
22: $\mathbf{G}_l \leftarrow \text{zeros}\big(\max_j \max(\{x_1\colon (x_1, x_2) \in \text{dom }\hat{f}_j^{(l)}\}), n_l, \max_j \max(\{x_2\colon (x_1, x_2) \in \text{dom }\hat{f}_j^{(l)}\})\big)$
23: **for** $j = 1$ to $n_l$ **do**
24:     **for** $(x_1, x_2)$ in dom $\hat{f}_j^{(l)}$ **do**
25:         $G_l(x_1,\, j,\, x_2) \leftarrow \hat{f}_j^{(l)}(x_1, x_2)$
26:     **end for**
27: **end for**
28: Return cores $\mathbf{G}_1, \ldots, \mathbf{G}_d$.

---

This matrix can be written in block form as

$$\mathbf{G}(:,\, 3,\, :) = \begin{pmatrix} O_1 & O_2 \\ I & O_3 \end{pmatrix},$$

where $O_1$, $O_2$ and $O_3$ are zero matrices, and $I$ is the identity matrix. If the result of the calculation of the previous function is 3, which corresponds to the vector $(e^{(4)})^T$ as the result of multiplication of all previous kernels, then after multiplying by the slice data we obtain $(e^{(6)})^T = (e^{(4)})^T \mathbf{G}(:,\, 3,\, :)$. The basis vector $(e^{(6)})^T$ expectedly corresponds to the value of the function equal to 5.

But in the more complex case, the correspondence between the function value and its vector representation may not be obvious. Consider the following first two vectors

$$\boldsymbol{a}_1 = \{1,\, 2,\, 10\}, \quad \boldsymbol{a}_2 = \{-1,\, 0,\, 5,\, 8\}.$$

Then the second slice of the second core $\mathbf{G}_2(:,\, 2,\, :)$ which is generated by the function $f(x) = x + 0$, is equal to

$$\mathbf{G}_2(:,\, 2,\, :) = \begin{pmatrix} 0 & 1 & 0 & 0 & 0 & 0 & 0 & 0 & 0 & 0 \\ 0 & 0 & 1 & 0 & 0 & 0 & 0 & 0 & 0 & 0 \\ 0 & 0 & 0 & 0 & 0 & 0 & 0 & 1 & 0 & 0 \end{pmatrix}.$$

This is true because the first derivative function has three values, which correspond to the first three basis vectors:

$$1 \Longleftrightarrow e^{(1)}, \ \ 2 \Longleftrightarrow e^{(2)}, \ \ 10 \Longleftrightarrow e^{(3)},$$

whereas the value area of the second derivative function has 8 elements, which correspond to

$$0 \Longleftrightarrow e^{(1)}, \ 1 \Longleftrightarrow e^{(2)}, \ 2 \Longleftrightarrow e^{(3)}, \ \ \ldots, \ \ 10 \Longleftrightarrow e^{(7)}, \ 15 \Longleftrightarrow e^{(8)}, \ 18 \Longleftrightarrow e^{(9)}.$$

Note that in the examples above we chose to assign basis vectors to the values of the derived functions according to their their ascending sorting. This is not a crucial point, since this correspondence is conditional, it can change and be different for each core.

In the degenerate case, when the function $P$ is identical: $P(x) = x$ we can construct the desired TT-decomposition with ranks equal to 2. Namely, in this case the cores have the following explicit form

$$\mathbf{G}_1(:, i, :) = (1, \ a_1[i]); \quad \mathbf{G}_k(:, i, :) = \begin{pmatrix} 1 & a_k[i] \\ 0 & 1 \end{pmatrix}, \ \ 2 \le k \le d-1; \quad \mathbf{G}_d(:, i, :) = \begin{pmatrix} a_d[i] \\ 1 \end{pmatrix}. \tag{9}$$

These cores can be constructed using our techniques as follows. Consider the following tensor with binary indices $i_k \in \{0, 1\}$ which is equal to 1 iff only one of its indices is 1:

$$I(i_1, i_2, \ldots, i_d) = \begin{cases} 1, & \sum_{k=1}^{d} i_k = 1, \\ 0, & \text{else} \end{cases}.$$

We can construct its cores $\widehat{\mathbf{G}}$ using the following derivative functions

$$f_0^k(x) = x, \quad f_1^k(x) = \begin{cases} 1, & x = 0, \\ \text{None}, & \text{else}. \end{cases}$$

And consider one-rank tensor $\mathbf{H}$ with values to be summed up with the following cores

$$\mathbf{K}_k(:, 0, :) = (1), \quad \mathbf{K}_k(:, 1, :) = (v_k).$$

For this tensor it is true that

$$H(\underbrace{0, 0, \ldots, 0}_{k-1}, 1, 0, \ldots, 0) = v_k,$$

thus its convolution with the tensor $\mathbf{I}$ gives the sum of elements of $v$:

$$\sum_{i_1=0}^{1} \sum_{i_2=0}^{1} \cdots \sum_{i_d=0}^{1} I(i_1, i_2, \ldots, i_d) H(i_1, i_2, \ldots, i_d) = \sum_{k=1}^{d} v_k. \tag{10}$$

This convolution operation is shown schematically in Fig. 5a.

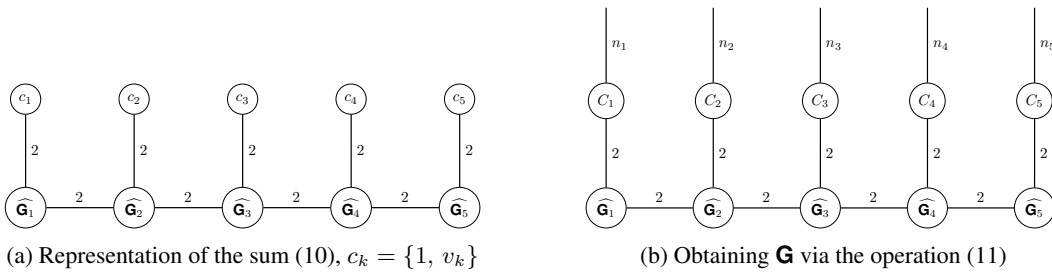

(a) Representation of the sum (10), $c_k = \{1, v_k\}$      (b) Obtaining $\mathbf{G}$ via the operation (11)

Figure 5: Tensor network for TT-tensor for simple sum for $d = 5$.

Now, to be able to select the terms by choosing the indices of the tensor $\mathbf{G}$, we replace the vectors $a$ in this network with matrices $C_k$ of the following form

$$C_k = \begin{pmatrix} 1 & 1 & 1 & \cdots & 1 & 1 \\ v_k[1] & v_k[2] & v_k[3] & \cdots & v_k[n_k - 1] & v_k[n_k] \end{pmatrix},$$

where $n_k$ is the length of the vector $\boldsymbol{a}_k$. Finally, we obtain the cores $\mathbf{G}_k$ by the convolution of the cores $\widehat{\mathbf{G}}_k$ and the matrices $C_k$ (see Fig. 5b.):

$$\mathbf{G}_k(:, i, :) = \sum_{j=0}^{1} \widehat{\mathbf{G}}(:, j, :) C_k(j, i). \tag{11}$$

**Step function in QTT format**  The definition of step function in QTT format and its derivative functions are given in Section 2. Let us consider in more detail how these functions give the desired value of the tensor built on them.

Further, we assume that the number $t$ appearing in the definition of the step function is constant, and that the multi-index $\{i_1, i_2, \ldots, i_d\}$ we choose to obtain a particular tensor value may be different. Recall, that $t$ has the following bit decomposition

$$t = \sum_{j=0}^{d-1} b_{d-i} 2^j.$$

Multi-index $\{i_1, i_2, \ldots, i_d\}$ have the meaning of bit decomposition of the argument $x$ of the step function

$$x = \sum_{j=0}^{d-1} i_{d-i} 2^j.$$

Thus, if $x \leq t$ then the tensor must return 0, otherwise its value is equal to 1.

Recall, that in this example derivative functions are the following:

If $b_k = 0$, then $f_0^k(x) := f^{(k)}(0, x) = x$,        $f_1^k(x) := f^{(k)}(1, x) = 1$;

if $b_k = 1$, then $f_0^k(x) := f^{(k)}(0, x) = \begin{cases} 1, & x = 1 \\ \text{None}, & x = 0. \end{cases}$,        $f_1^k(x) := f^{(k)}(1, x) = x$.

Assume, that for some $k$, $1 \leq k < d$ all bit in the numbers of $t$ and $x$ before this index are coincide:

$$b_j = i_j, \quad 1 \leq j < k$$

and their $k$-th bits are different.

As the first case consider the case when $b_k = 0$ but $i_k = 1$. It means that $x > t$ and the value of the tensor must be 1: $\mathsf{l} = 1$, regardless of the correlation between subsequent bits, as they correspond to the lower bits. Let's trace what values the derivative functions receives as input and outputs to get the value of 1 at the end. For bits with indices less than $k$, the derivative functions are equal to the identity function. Indeed, if for some $1 \leq j < k$, $b_j = 0$, it means, that $i_j = 0$ and derivative function corresponding to this sequence member is taken from the first line of the definition above: $f^{(j)}(0, x) = x$. If for some $1 \leq j < k$, $b_j = 1$, then derivative function corresponding to this sequence member is taken from the second line of the definition above: $f^{(j)}(1, x) = x$. Thus, the $k$-th derivative function receives as input the initial value that is fed to the input of the very first function and in this case this value is not essential. The $k$-th derivative function, in turn, returns 1 according to its definition in the first line of the expression with definitions of $f$: $f^{(k)}(1, x) = 1$. This value will be "carried" to the end without change as for all derivative functions in this example equality $f(1) = 1$ holds. It gives the resulting value 1 of the TT-tensor. So we have achieved the expected behaviour in this case.

If, on the contrary, $b_k = 1$ and $i_k = 0$ then we surely know that $\mathsf{l} = P(x) = 0$ as $x < t$ in this case. As we discussed in the previous case, the $k$-th derivative function receives an initial value as input. In this case it is important that this value is equal to 0. The $k$-th derivative function is undefined at this value of the argument, which we symbolically write as $f_k^{(0)}(0, 0) = \text{None}$. All derivative functions are pre-defined in such a way that they always return None if they are not an middle-function, and the middle-function returns 0 if it has at least one argument that is None. In our example, the middle-function is the last ($d$-th) function, and therefore has two arguments. Thus, the None value will be "carry on" until the last function, which will return 0. In the language of

vectors, this means that after multiplication by the $k$-th core we get a zero vector, not a basis vector, so that for any subsequent values of the indices the result will be zero. So we have achieved the expected behaviour in this case too.

In the remaining case when $b_j = i_j$ for all $1 \leq j \leq d$, the derivative functions leave the initial argument unchanged. This argument is 0. Thus, the constructed TT-tensor have the expected value in this case as well.

The maximum TT-rank value in this example is no more than 2. For the case of rank equal to 2, the explicit form of the cores are the following, for $b_k = 0$:

$$\mathbf{G}_k(:, 0, :) = \begin{pmatrix} 1 & 0 \\ 0 & 1 \end{pmatrix}, \quad \mathbf{G}_k(:, 1, :) = \begin{pmatrix} 0 & 1 \\ 0 & 1 \end{pmatrix}, \quad b_k = 0,$$

and for $b_k = 1$:

$$\mathbf{G}_k(:, 0, :) = \begin{pmatrix} 0 & 0 \\ 0 & 1 \end{pmatrix}, \quad \mathbf{G}_k(:, 1, :) = \begin{pmatrix} 1 & 0 \\ 0 & 1 \end{pmatrix}, \quad b_k = 1.$$

By direct calculations it can be seen that if we assign to number 0 the basis vector $e^{(1)} = \{1, 0\}$, and to number 1 the basis vector $e^{(2)} = \{0, 1\}$, then multiplication of these vectors by the prescribed matrices gives exactly the same results as the application of derivative functions for this example.

## B.2 COOPERATIVE GAMES

In this subsection, we take a closer look at the examples that were briefly given in Section 3.1, and we also give some more examples of applying our methods to cooperative games, which are described in (Ballester-Ripoll, 2022).

**Shoe sale game**  Brief game conditions and the derivative functions for this game are described in general form above. Consider what happens when a certain number $k$ is given, i. e. the order number of the player for whom we calculate the payoff $\pi(k)$. For each $k$ we rebuild tensors $p(|\mathbb{S}|)\nu(\mathbb{S} \cup \{k\})$ and $p(|\mathbb{S}|)\nu(\mathbb{S})$. At the $k$-th place we leave only one function of the two: $f_1^{(k)}$ for the first specified tensor, and $f_0^{(k)}$ for the second. Thus, although we formally have a tensor of dimension $d$, there is actually no summation at the $k$-th index, and we obtain a sum over all subsets of the set $\mathbb{T} \setminus \{k\}$. In addition, in the case of the tensor $p(|\mathbb{S}|)\nu(\mathbb{S} \cup \{k\})$, we subtract a unit from the argument $p$ since in this case $p(|\mathbb{S}|)\nu(\mathbb{S} \cup \{k\}) = p(|\mathbb{S} \cup \{k\}| - 1)\nu(\mathbb{S} \cup \{k\})$.

When we go through the computational tree of Figure 2a using the derivative functions of this example, we have the following. Let $k > L + 1$ and multi-index $\{i_1, i_2, \ldots, i_d\}$ has the meaning given in the main article: if any index $i_j$ equals 1, it means that the player with number $j$ is part of the coalition, if 0, he is not. When we go from the left end, each derivative function adds 1 to its argument if the corresponding index is 1. If the corresponding player is not part of the coalition, the corresponding index is zero and the derivative function returns the argument unchanged. The "left" functions thus count the number of players in the coalition.

When passing from the right end the same happens, the number of players with numbers greater than $L + 1$ ("right players" in terms of this cooperative game) is counted, except for the player with the number $k$. This player is always counted when we build cores for the tensor $p(|\mathbb{S}|)\nu(\mathbb{S} \cup \{k\})$ and is not counted is the case of tensor $p(|\mathbb{S}|)\nu(\mathbb{S})$ as explained above.

Thus the middle-function with index $L + 1$ gets two arguments: $x$—the sum of the number of "left" players and $y$—the sum of the number of "right" players, excluding the player with the number $L+1$.

The middle-function adds one to $y$ if index $i_{L+1}$ is 1, and does not change it otherwise. Finally, the middle-function returns $\min(x, y)p(x + y)$ which coincides with the expected value of the tensor.

At the next step we contract each tensor with ones, namely, we calculate the following expression

$$\left( \sum_{i_1=1}^{n_1} \mathbf{G}_1(1, i_1, :) \right) \left( \sum_{i_2=1}^{n_2} \mathbf{G}_2(:, i_2, :) \right) \cdots \left( \sum_{i_d=1}^{n_d} \mathbf{G}_d(:, i_d, 1) \right) \tag{12}$$

from left to right in sequence. In the case of this example, all $n_j = 2$, $j = 1, \ldots, d$. Thus we build and contract $2N = 4L + 2$ tensors, but since we do not build full cores in memory and instead work in a compressed format, we get the computation times shown in Fig. 3a.

**Airport**   Brief game conditions and the derivative functions for this game are described in general form above.

In this problem we once explicitly build the cores for the tensor $\nu$, making a convolution with matrices $A_k$ (see Fig. 4a). This convolution in fact, is reduced to the replacement of the corresponding unit in the core slice by the value of $c_k$. Separately, we construct the cores for the tensor $p$, which is the given function of the sum of all indices equal to 1. The construction of such a tensor is described at the beginning of Sec. B.1; in this case we put $a_j[0] = 0$, $a_j[1] = 1$, $j = 1, \ldots, d$.

To calculate the value of $\pi(k)$, we applied a trick similar to the one used in (Ballester-Ripoll, 2022). Namely, for each $k$ we have left the core $\mathbf{G}_k^{(p)}$ from the TT-decomposition to the tensor $p$ only the first slice of dimensionality, which corresponds to the index $i_k = 0$, so the following is true for the new core $\widehat{\mathbf{G}}_k^{(p)}$:

$$\widehat{\mathbf{G}}_k^{(p)}(:, 0, :) = \mathbf{G}_k^{(p)}(:, 0, :), \quad \widehat{\mathbf{G}}_k^{(p)} \in \mathbb{R}^{r_{k-1}^{(p)} \times 1 \times r_k^{(p)}}.$$

For the core $\mathbf{G}_k^{(\nu)}$ from the TT-decomposition of the tensor $\nu$ we also removed the second slice and took the slice difference of the original core as the first slice:

$$\widehat{\mathbf{G}}_k^{(\nu)}(:, 0, :) = \mathbf{G}_k^{(\nu)}(:, 1, :) - \mathbf{G}_k^{(\nu)}(:, 0, :), \quad \widehat{\mathbf{G}}_k^{(\nu)} \in \mathbb{R}^{r_{k-1}^{(\nu)} \times 1 \times r_k^{(\nu)}}.$$

For the final result, we have taken a convolution of these tensors, which in TT-format is written as

$$\pi(k) = \left( \sum_{i_1=1}^{2} \mathbf{G}_1^{(p)}(1, i_1, :) \otimes \mathbf{G}_1^{(\nu)}(1, i_1, :) \right) \left( \sum_{i_2=1}^{2} \mathbf{G}_2^{(p)}(:, i_2, :) \otimes \mathbf{G}_2^{(\nu)}(:, i_2, :) \right) \cdots$$
$$\cdots \mathbf{G}_k^{(p)}(:, 1, :) \otimes \mathbf{G}_k^{(\nu)}(:, 1, :) \cdots \left( \sum_{i_d=1}^{2} \mathbf{G}_d^{(p)}(:, i_d, 1) \otimes \mathbf{G}_d^{(\nu)}(:, i_d, 1) \right),$$

where $\otimes$ denotes the Kronecker product of matrices.

For the numerical experiments we take values $c_k$ as i.i.d. random values uniformly distributed on the interval $[0, 1]$.

**Weighted majority game**   For this game, briefly described in Section 3.1, we took the following derivative functions for the tensor $p(|\mathbb{S}|)(\nu(\mathbb{S} \cup \{k\}) - \nu(\mathbb{S}))$ of dimension $d = |\mathbb{T}| - 1$:

$$f_0^j(x) = x, \quad f_1^j(x) = \begin{cases} \{x[1] + w_j, \, x[2] + 1\}, & x[1] + w_j \leq M, \\ \text{None}, & \text{else} \end{cases}, \quad 1 \leq j < k,$$

$$f_0^j(x) = x, \quad f_1^j(x) = \begin{cases} \{x[1] + w_{j+1}, \, x[2] + 1\}, & x[1] + w_{j+1} \leq M, \\ \text{None}, & \text{else} \end{cases}, \quad k \leq j < d,$$

and for the middle-function, which is the last one in this example, we have

$$f_i^d(x) = \begin{cases} p(x[2] + i), & x[1] + iw_{d+1} + w_k \geq M > x[1] + iw_{d+1}, \\ \text{None}, & \text{else} \end{cases}, \quad i = 0, 1.$$

(In the case of $k = |\mathbb{T}| = d + 1$, we take $w_d$ instead of $w_{d+1}$). Note that in this example the derivative functions are defined on a set of vectors of length 2. The first component of this vector accumulates a sum of weights $\{w_j\}$ to compare with the threshold $M$. The second component of the input vector counts the number of players in the coalition $\mathbf{S}$, this value is passed to the function $p$ in the last derivative function.

If at step $j$, $j < d$ it turns out that the accumulated sum of weights already exceeds the threshold $M$ ($x[1] + w_j > M$) then the derivative function returns $\texttt{None}$, thus zeroing out the value of the tensor.

Indeed, in this case, the difference $\nu(\mathbb{S} \cup \{k\}) - \nu(\mathbb{S}))$ will be obviously equal to zero regardless of what other coalition members are added, i.e. regardless of $i_l$, $l > j$ index values. This trick reduces the TT-ranks of the resulting tensors and thus reduces the execution time.

In this example, we construct the cores of the specified tensor $N = |\mathbb{T}|$ times, and then perform its convolution by the formula (12) in the sparse format. For the numerical experiments we take values $\{w_k\}$ as i.i.d. random integers uniformly distributed on the interval $[1, 10]$ and take threshold equal to $M = \lfloor 1/2 \sum_k w_k \rfloor + 1$.

Technically, we can avoid passing vectors to the deriving functions, but limit ourselves to an integer argument $X$ equal to $X = x[1] + x[2]N_{\text{big}}$, where $N_{\text{big}}$ is a sufficiently large integer ($N_{\text{big}} = 2^{15}$). Then in each derivative function we produce an "unpacking" $x[1] = X \mod N_{\text{big}}$ and $x[2] = \lfloor X/N_{\text{big}} \rfloor$.

**Bankruptcy**  This game has a function of $\max$ on the sum of the values, so it is mathematically similar to the airport game, but we have chosen a different way of constructing derivative functions. Namely, we take

$$f_0^j(x) = \begin{cases} \{x[1] - c_{\text{add}}(i), \, x[2]\}, & x[1] - c_{\text{add}}(i) > 0, \\ \text{None}, & \text{else} \end{cases}, \quad f_1^j(x) = \{x[1], \, x[2] + 1\}, \quad 1 \le j < d,$$

where $c_{\text{add}}(i) = c[i]$ if $i < k$ and $c_{\text{add}}(i) = c[i+1]$, else. For the middle function, which is the last one ($l = d = N - 1$), we have:

$$f_0^d(x) = \begin{cases} (x[1] - c_{\text{add}} - \max(0, \, x[1] - c_{\text{add}} - c[k]))p(x[2]), & x[1] - a_{\text{add}} > 0, \\ 0, & \text{else} \end{cases},$$

where $c_{\text{add}} = c[d+1]$ if $k < d+1$, and $c_{\text{add}} = c[d]$ if $k = d+1$;

$$f_1^d(x) = \big(x[1] - \max(0, \, x[1] - c[k])\big)p(x[2] + 1).$$

Note that in the case of this example, the initial functions $f_1$ are given the value $\{E, 0\}$ as input instead of $0$. This is done to simplify the type of derivative functions. Like in Weighted majority game, derivative functions take as input a vector of 2 elements, the first of which accumulates the values of $\{c_j\}$, and the second element summarizes the number of players in a coalition.

In this example, we construct the cores of the specified tensor $N = |\mathbb{T}|$ times, and then perform its convolution by the formula (12) in the sparse format. For the numerical experiments we take values $\{c_k\}$ as i.i.d. random integers uniformly distributed on the interval $[1, 10]$ and take the value of $E$ equal to $E = 1/2 \sum_k c_k$.

**One seller market game**  In this game, first player in selling some good, players $2, \ldots, |\mathbb{T}|$ offer prices $\{a_2, a_3, \ldots, a_{|\mathbb{T}|}\}$ for this good, $a_k \ge 0$. If the first player is in a coalition $\mathbb{S}$, the value of this coalition is equal to the maximum price offered by the members of this coalition

$$\nu(\mathbb{S}) = \max_{k \in \mathbb{S}, \, k \ne 1} a_k.$$

If there is no first player in the coalition, its price is zero: $\nu(\mathbb{S}) = 0$.

For the first player we take the following derivative functions

$$f_0^1(x) = \text{None}, \quad f_1^1(x) = 0,$$

and for the rest of the players:

$$f_0^k(x) = x, \quad f_1^k(x) = \max(x, a_k).$$

The problem under consideration is another example of such a tensor construction, the ranks of which depend on the order of the indices. For an unsuccessful sequence, the TT-ranks of the resulting tensor can be large and it is necessary to perform an SVD-step for reducing the TT-ranks. The final ranks depend on specific values of $\{a_k\}$.

To reduce the ranks, we we can change the sequence of players, as it will not affect the calculation of the sum (8). Namely, place the first player first, and sort the other players according to their pre-determined prices in descending order. With this sorting we can take the same derivative function for $k > 1$ as in Airport game.

Thus, in this problem we come to the tensor network shown in Fig. 6 with matrices $\boldsymbol{A}_k$ of the same kind as the matrices in the airport problem with the following elements

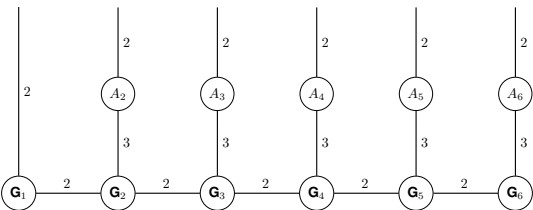

Figure 6: Tensor network for building TT-tensor for the one seller market game with optimal players ordering for $|\mathbb{T}| = 6$. The numbers near lines denote ranks (number of terms in sum).

$$\boldsymbol{A}_k = \begin{pmatrix} 1 & 0 & 0 \\ 0 & 1 & a_k \end{pmatrix}, \quad k = 2, \ldots, |\mathbb{T}|.$$

Technically, the convolution procedure can be omitted explicitly, taking it into account at the stage of tensor construction.

Note that since the cores **G** of the resulting TT-tensor are constructed using the same pattern as well as the cores of the tensor representing values of $p$, they can be built on the fly rather than stored when calculating the sum (8). Moreover, we can abandon the construction of the cores at all, immediately producing all algebraic operations, given the special (sparse) kind of kernels. For the considered problem, the sequence of matrix operations to calculate the sum (8) is reduced to the Algorithm 3.

---

**Algorithm 3** Algorithm for calculating the sum (8)for one seller market game, based on obtained TT-cores

---

**Require:** Values $\{a_i\}$ sorted in descending order, number $k$, weight function $P$
**Ensure:** Value of $\pi_k$
  1: $v \leftarrow \underbrace{\{0, 0, \ldots, 0\}}_{2|\mathbb{T}|}$ # *Initialization*
  2: $v[0 : 4] \leftarrow \{0, 1\} \otimes \{1, 0\}$
  3: **for** $i = 1$ to $|\mathbb{T}| - 2$ **do**
  4:     **if** $i == k$ **then**
  5:         **for** $j = i$ to 0 step $-1$ **do**
  6:             $v[2j + 1] \leftarrow a_{i+1} \cdot v[2j]$
  7:             $v[2j] \leftarrow -v[2j]$
  8:         **end for**
  9:     **else**
 10:         **for** $j = i$ to 0 step $-1$ **do**
 11:             $v[3 + 2j + 1] \leftarrow v[3 + 2j + 1] + a_{i+1} \cdot v[2j] + v[2j + 1]$
 12:         **end for**
 13:     **end if**
 14: **end for**
 15: $s \leftarrow 0$
 16: **for** $i = 1$ to $|\mathbb{T}| - 2$ **do**
 17:     $s \leftarrow s + v[2i]P(i + 1)a_{|\mathbb{T}|}$
 18:     $s \leftarrow s + v[2i + 1](P(i) + P(i + 1))$
 19: **end for**
 20: $s \leftarrow s + v[2|\mathbb{T}| - 1]P(|\mathbb{T}| - 1)$
 21: Return $s$

---

This algorithm works only for $1 < k < |\mathbb{T}|$ and it is not the most efficient. However, it is shown to illustrate the possibility of applying the technique described in the article to build this kind of efficient iterative algorithms. It is worth noting that the complexity of this algorithm is $O(|\mathbb{T}|^2)$.

### B.3 KNAPSACK PROBLEM

Consider a knapsack problem. The formulation is the following: we have $n$ type of items, each item have weight $w_i \geq 0$ and value $v_i \geq 0$, $i = 1, 2, \ldots, n$. The task is to solve optimization problem

$$\underset{\{x_1, x_2, \ldots, x_n\}}{\text{maximize}} \sum_{i=1}^{n} v_i x_i, \quad \text{s.t.} \sum_{i=1}^{n} w_i x_i \leq W,$$

where $\{x_i\}_{i=1}^{n}$ are the unknown number of each item, value $W \in \mathbb{R}$ is the given maximum total weight. By imposing different constraints on the unknowns $x_i$, we obtain different formulations of the problem.

**0–1 knapsack problem** First, consider a formulation in which it is assumed that $x_i \in \{0, 1\}$ (so called *0–1 knapsack problem*). To solve this problem, we construct two tensors. First tensor **V** represent the total cost of the knapsack

$$V(x_1, x_2, \ldots, x_n) = \sum_{i=1}^{d} v_i x_i.$$

Since the expression on the right side of this definition is linear, this tensor can be represented in TT-format with TT-ranks of no more than 2 (see (9)). Namely, the cores $\{\mathbf{H}_k\}$ of the TT-decomposition of the tensor **V** are

$$\mathbf{H_1}(:, 0, :) = \begin{pmatrix} 1, & 0 \end{pmatrix}; \quad \mathbf{H_k}(:, 0, :) = \begin{pmatrix} 1 & 0 \\ 0 & 1 \end{pmatrix}, \quad 2 \leq k \leq d-1; \quad \mathbf{H_d}(:, 0, :) = \begin{pmatrix} 1 \\ 0 \end{pmatrix};$$

$$\mathbf{H_1}(:, 1, :) = \begin{pmatrix} 1, & v_1 \end{pmatrix}; \quad \mathbf{H_k}(:, 1, :) \begin{pmatrix} 1 & v_k \\ 0 & 1 \end{pmatrix}, \quad 2 \leq k \leq d-1; \quad \mathbf{H_d}(:, 1, :) = \begin{pmatrix} v_d \\ 1 \end{pmatrix}.$$

Here we use indices 0 and 1 for the middle indices of the TT-cores so that they correspond to the physical sense of the task—the presence or absence of this item in the knapsack.

The second tensor **I** is the indicator tensor of the condition in the knapsack problem:

$$I(x_1, x_2, \ldots, x_n) = \begin{cases} 1, \text{ if } \sum_{i=1}^{n} w_i x_i \leq W, \\ 0, \text{ else.} \end{cases}$$

We can build functional TT-decomposition of the tensor **I** with the following set of functions

$$f_i^k(x) = \begin{cases} x + iw[k], & x + iw[k] \leq W, \\ \texttt{None}, & \text{else} \end{cases}, \quad i = 0, 1; \ 1 \leq k \leq d.$$

Note that the condition of not exceeding the weight is checked in each function, i. e., the conditions for partial sums are checked: $w_1 x_1 \leq W$, $w_1 x_1 + w_2 x_2 \leq W$, etc. This does not affect the final result as $w_i \geq 0$, but it allows us to reduce the ranks of the cores. The TT-ranks in this problem are highly dependent on the specific weights of the knapsack elements: whether they are integer, how large a range of values they have, etc.

The final answer to the knapsack problem is given by finding the maximum of the tensor, which is the elementwise product of the constructed tensors

$$\underset{\{x_1, x_2, \ldots, x_n\}}{\arg\max} V(x_1, x_2, \ldots, x_n) \cdot I(x_1, x_2, \ldots, x_n).$$

The TT-cores of such a tensor are found as the Kronecker product of the slices of the multiplier cores (see (Oseledets, 2011)). The operation of approximate finding the maximum value of the tensor in TT-format is implemented, for example, in the package `ttpy`.

**Multi-Dimensional bounded knapsack problem**   Another addition, which can be implemented with slight modifications of the above scheme, is related to the presence of several constraints. Namely, let there be several weights $\{w_i^{(j)}\}$ associated with each item, with separate conditions imposed on them

$$\underset{\{x_1, x_2, \ldots, x_n\}}{\text{maximize}} \sum_{i=1}^{n} v_i x_i, \quad \text{s.t.} \sum_{i=1}^{n} w_i^{(j)} x_i \leq W^{(j)}, \quad j = 1, \ldots, M; \quad x_i = 0, 1, \ldots N_i,$$

where $M$ is the length of condition vector.

To solve this problem, we generate $M$ indicator tensors $\{\mathbf{I}^{(j)}\}_{j=1}^{M}$, one for each condition, according to the algorithm above. The we find element-wise product of all this tensors, and find (quasi-) maximum element in the resulting tensor

$$\underset{\{x_1, x_2, \ldots, x_n\}}{\arg\max} \mathbf{V}(x_1, x_2, \ldots, x_n) \cdot \prod_{j=1}^{M} \mathbf{I}^{(j)}(x_1, x_2, \ldots, x_n).$$

The product of a large number of tensors $\mathbf{I}^{(j)}$ can lead to a rapid growth of ranks, to get around this we can round the tensor after each successive multiplication.

### B.4   Partition problem

Consider the partition problem in the following formulation. We have a multiset $\mathbb{S}$, $|\mathbb{S}| = d$ of positive integers (multiset is a set with possibly repeating elements) and an integer $n$. The task is to partition the set $\mathbb{S}$ on $n$ sets $\{\mathbb{S}_i\}_{i=1}^{n}$ such that the sum of elements in each set are equal:

$$\sum_{a \in \mathbb{S}_1} a = \sum_{a \in \mathbb{S}_2} a = \cdots = \sum_{a \in \mathbb{S}_n} a = \frac{1}{n} \sum_{a \in \mathbb{S}} a.$$

In order to use our approach to solve this problem, we construct $n$ indicator tensors, each of which corresponds to one of the equalities in the expression above. Namely, the $j$-th tensor is defined as

$$\mathbf{I}_j(i_1, i_2, \ldots, i_d) = \begin{cases} 1, \text{ if } \sum_{k=1}^{d} s[k] \cdot \delta(i_k, j) = T, \\ 0, \text{ else} \end{cases},$$

where $s[k]$ is the $k$-th element of the (ordered in some way) set $\mathbb{S}$, $\delta$ is the Dirac delta function and $T := \frac{1}{n} \sum_{a \in S} a$.

The maximum indices value of this tensor is $n$. The index value $i_k = l$ means that the $k$-th element of the set $\mathbb{S}$ belongs to the set $\mathbb{S}_l$. Thus we have one-to-one correspondence between indices set $\{i_1, i_2, \ldots, i_d\}$ and a partition of the set $\mathbb{S}$.

Derivative functions for the $j$-th tensor are the following

$$f_i^k(x) = \begin{cases} x + s[k], \text{ if } i = j, \\ x, \text{ else} \end{cases}, \quad 1 \leq k < d, \ 1 \leq i \leq n,$$

and the middle function which is the last one:

$$f_i^d(x) = \begin{cases} 1, & \text{if } x + s[d] = T \text{ and } i = j, \\ 1, & \text{if } x = T \text{ and } i \neq j, , \quad 1 \leq i \leq n. \\ 0, & \text{else} \end{cases}$$

Finally we construct indicator tensor $\mathbf{I}$ as a Hadamard product of the built tensors

$$\mathbf{I} = \mathbf{I}_1 \odot \mathbf{I}_2 \odot \cdots \odot \mathbf{I}_n.$$

The value of this tensor $\mathbf{I}$ is 1 only for the indices corresponding to the desired problem statement. We can find them by finding the maximal element of the indicator TT-tensor, see Section D in the Appendix.

```
def f(x, i):
    last = i == N-1
    if (x[i]) > 0 or \
       (x[i + N]) > 0 or \
       (x[i + 2*N]) > 0:
        return None

    if last: # middle-core
        return 1

    # add our position
    # to all sets
    x[i] = 1
    x[i + N] = 1
    x[i + 2*N] = 1

    # shift bottom-to-top set
    x[N:2*N-1] = x[N+1:2*N]
    x[2*N-1] = 0

    # shift top-to-bottom set
    x[2*N+1:3*N] = x[2*N:3*N-1]
    x[2*N] = 0

    return x
```

(a) derivative functions in Python

(b) Positions for $N = 9$ and 10.

Figure 7: Eight queens puzzle

## B.5 EIGHT QUEENS PUZZLE

Consider the classical problem of eight queens and its extensions. In the classical variant it is necessary to place 8 queens on a usual $8 \times 8$ chessboard so that the queens do not beat each other. In other words, no two queens stand on the same vertical, horizontal, or diagonal. We will consider this problem on an $N \times N$ board with $N$ queens. To solve this problem by our techniques we construct a tensor of dimension $N$, each $k$-th index $i_k$ of which denote the position of the queen on the $k$-th vertical, so $1 \leq i_k \leq N$. The value of the tensor is an indicator function of the desired state: 1 if the location of the queens satisfies the condition and zero otherwise.

The derivative functions $f_k^i(x)$ in Python for such a tensor are shown in Fig. 7a. The are the same for all $k$, middle-core is the last core, the above code also covers the function for the last core.

Let us briefly describe their work. The input is not a single number, but an array (it is allowed in our scheme) of zeros and ones. The first $N$ bits show the position of the previous (leftmost) queens horizontally. The bit corresponding to the current queen is added to them. The next $N$ bits show those fields which are broken from bottom to top, and they are shifted forward. And finally, the last series of $N$ bits shows those fields that are broken from top to bottom, and they are shifted back. If the position of the bit corresponding to the position of the current queen contains a one in at least one of the sets, the producing function returns `None`, since it means that the condition has already been violated.

After constructing the tensor, we can find the desired position: this problem is reduced to finding a non-zero element in the tensor (in this case the value of 1), and its algorithm is described in the Appendix, Section D. The result is shown on Fig. 7b.

Note that although the state that is passed to each derivative function is $3N$ in length and thus can potentially take $2^{3N}$ combinations, the real TT-rank is much smaller, since only a small fraction of this set of combinations is admissible. See Table 1 for the numerical values for the ranks together with the error in calculating the total number of combinations

| $N$ and truncation threshold $\epsilon$ | TT-ranks | # of positions |
|---|---|---|
| 8, w/o truncation | 1—8—42—140—339—538—482—224—1 | 92 |
| 8, $\epsilon = 0$ | 1—8—36—62—74—62—36—8—1 | $92.0 \pm 2 \cdot 10^{-14}$ |
| 8, $\epsilon = 10^{-6}$ | 1—8—36—62—74—62—36—8—1 | $92.0 \pm 0.0$ |
| 9, w/o truncation | 1—9—56—234—726—1565—2153—1734—740—1 | 352 |
| 9, $\epsilon = 0$ | 1—9—56—221—592—712—191—54—9—1 | $352.0 \pm 2 \cdot 10^{-13}$ |
| 9, $\epsilon = 10^{-6}$ | 1—9—54—172—246—246—172—54—9—1 | $352.0 \pm 4 \cdot 10^{-13}$ |
| 10, w/o truncation | 1—10—72—364—1393—3842—7289—8838—6426—2576—1 | 724 |
| 10, $\epsilon = 0$ | 1—10—72—339—914—1225—1820—391—72—10—1 | $724.0 \pm 5 \cdot 10^{-13}$ |
| 10, $\epsilon = 10^{-6}$ | 1—10—72—284—526—606—526—284—72—10—1 | $724.0 \pm 0.0$ |

Table 1: TT-ranks and the number of positions calculated using the tensor depending on the size of the boar $N$ and the truncation threshold $\epsilon$.

Of course, solving the problem of finding permissible combinations in the described way is inefficient. But we can get the number of possible combinations even faster than finding one of them. This number is obtained by convolution of the tensor with vectors consisting of ones. Besides, by constructing cores of decomposition, we can solve extended problems. For example, the complement problem, when some of the queens are already placed on the board, and we need to find the location of the others. Or we can consider that queens are placed on the board randomly, with a given probability distribution law, and the problem is to find the probability that such an arrangement will result in the position given by the puzzle rules.

### B.6 COMBINATORIAL PROBLEMS

**Sawtooth sequence** Sawtooth sequence is a sequence of integer numbers $\{a_1, a_2, \ldots, a_d\}$ such that if $a_{i-1} < a_i$ then $a_i > a_{i+1}$, and vice versa.

Suppose we have a set of arrays $\{c_j\}$, and for a given set of indices $i_1, i_2, \ldots, i_d$ we want to construct an indicator tensor equal to 1 if the corresponding sequence $\{c_1[i_1], c_2[i_2], \ldots, c_d[i_d]\}$ is sawtooth.

In this example, the derivative functions receive as input a sequence of two elements, the first element contains the value of the previous member of the sequence, and the second denotes the direction, "up" or "down".

$$f_i^k(x) = \begin{cases} \{c_k[i], \text{``}up''\}, & \text{if } c_k[i] < x[1] \text{ and } x[2] = \text{``}down'' \\ \{c_k[i], \text{``}down''\}, & \text{if } c_k[i] > x[1] \text{ and } x[2] = \text{``}up'' \\ \text{None}, & \text{else} \end{cases}.$$

Middle function is the last one and it is of the same form. We need to slightly alter this definition for the first function for $k = 1$.

Such sequences are often found in game problems. We can combine this indicator tensor with other conditions. For example, we can find the number of possible tooth sequences with certain conditions through convolution (12) of the resulting tensor with units.

**Number of subsets** Consider Problem #10 from the Advanced problems chapter of the book (Andreescu & Feng, 2002): Find the numbers of subsets of $\{1, \ldots, 2000\}$, the sum of whose elements is divisible by 5.

We can immediately construct an indicator tensor with binary indices, which equals 1 if the given subset (of indices with value 1) satisfies the condition of the problem. Namely, let us take the following derivative functions

$$f_i^k(x) = (x + ik) \mod 5, \quad 1 \le k < d, \; i = 0, 1,$$

and the middle function which is the last one in this example as

$$f_i^d(x) = \begin{cases} 1, & \text{if } (x + ik) \mod 5 = 0 \\ 0, & \text{else} \end{cases}.$$

However, for the specified number of elements of the sequence (2000), it can be a time-consuming task to convolve such a tensor even in a sparse format. Therefore, let us try to solve this problem analytically, using the explicit representation of the cores of this tensor.

Note that for the product of any five cores of this tensor, starting from the number that gives 1 when divided by 5, it is true

$$\left( \sum_{i_{5n+1}=1}^{2} \mathbf{G}_{5n+1}(:, i_{5n+1}, :) \right) \left( \sum_{i_{5n+2}=1}^{2} \mathbf{G}_{5n+2}(:, i_{5n+2}, :) \right) \left( \sum_{i_{5n+3}=1}^{2} \mathbf{G}_{5n+3}(:, i_{5n+3}, :) \right) \times$$

$$\times \left( \sum_{i_{5n+4}=1}^{2} \mathbf{G}_{5n+4}(:, i_{5n+4}, :) \right) \left( \sum_{i_{5n+5}=1}^{2} \mathbf{G}_{5n+5}(:, i_{5n+5}, :) \right) = \begin{pmatrix} 8 & 6 & 6 & 6 & 6 \\ 6 & 8 & 6 & 6 & 6 \\ 6 & 6 & 8 & 6 & 6 \\ 6 & 6 & 6 & 8 & 6 \\ 6 & 6 & 6 & 6 & 8 \end{pmatrix}.$$

Thus, to solve the problem it is necessary to power this matrix to 2000/5=400. To do this, let us write this symmetric matrix in the diagonal form:

$$\begin{pmatrix} 8 & 6 & 6 & 6 & 6 \\ 6 & 8 & 6 & 6 & 6 \\ 6 & 6 & 8 & 6 & 6 \\ 6 & 6 & 6 & 8 & 6 \\ 6 & 6 & 6 & 6 & 8 \end{pmatrix} = A \begin{pmatrix} 32 & 0 & 0 & 0 & 0 \\ 0 & 2 & 0 & 0 & 0 \\ 0 & 0 & 2 & 0 & 0 \\ 0 & 0 & 0 & 2 & 0 \\ 0 & 0 & 0 & 0 & 2 \end{pmatrix} A^T. \tag{13}$$

By direct calculations we find that for the matrix A and any $q, s \in \mathbb{R}$ we have

$$B := A \begin{pmatrix} q & 0 & 0 & 0 & 0 \\ 0 & s & 0 & 0 & 0 \\ 0 & 0 & s & 0 & 0 \\ 0 & 0 & 0 & s & 0 \\ 0 & 0 & 0 & 0 & s \end{pmatrix} A^T = \frac{1}{5} \begin{pmatrix} q+4s & q-s & q-s & q-s & q-s \\ q-s & q+4s & q-s & q-s & q-s \\ q-s & q-s & q+4s & q-s & q-s \\ q-s & q-s & q-s & q+4s & q-s \\ q-s & q-s & q-s & q-s & q+4s \end{pmatrix}.$$

Given that the first and last cores of the TT-disposition are vectors, not matrices, we need only the first element of the first row $B[1, 1] = 1/5(q + 4s)$. Using the diagonal form of the matrix (13) we can immediately power it to degree 400 by power its eigenvalues to this degree and obtaining $q = 32^{400}$ and $s = 2^{400}$.

Thus, the final answer to the problem: $1/5(32^{400} + 2^{400})$ number of subsets.

## B.7 SAT problem

Consider the standard Boolean satisfiability problem (SATisfiability) in conjunctive normal form (CNF-SAT). Given $d$ Boolean variables $\{x_i\}_{i=1}^d$, taking the value "True" or "False". From these variables we form $m$ logical expressions $\{A_i\}_{i=1}^m$ containing a set of some variables $\{x_i\}$ or their negations (which we denote by the symbol $\neg$) united by logical OR (symbol $\vee$), for example:

$$A_1 = x_1 \vee \neg x_3 \vee x_5, \quad A_2 = \neg x_1 \vee \neg x_2 \vee x_4 \vee x_5 \vee x_{10}, \quad \text{etc.}$$

The problem is to determine if there is such a set of variables $x$ that expressions $A_i$, combined with logical AND (symbol $\wedge$), give a logical "True":

$$A_1 \wedge A_2 \wedge \ldots \wedge A_m = \text{True}.$$

Let us use our method to construct a tensor with $n$ indices taking the values 0 and 1, corresponding to logical "True" and "False", which is equal to 1 if the latter equality is satisfied, and zero otherwise. Each index corresponds to a different variable $x_i$.

In this problem, we construct $m$ separate indicator tensors for each value $A_j$, which we then multiply elementwise, which corresponds to the logical AND operator. The indices of these tensors are binary and correspond to the `True` or `False` value of the corresponding variable. The derivative functions for the $j$-th tensor are the following:

$$f_{\text{True}}^k(x) = \begin{cases} x, & \text{if the variable } x_k \text{ is not part of the condition } A_j \\ x, & \text{if the variable } x_k \text{ is included in the condition } A_j \text{ with the negation} \\ 1, & \text{else} \end{cases},$$

$$f_{\text{False}}^k(x) = \begin{cases} x, & \text{if the variable } x_k \text{ is not part of the condition } A_j \\ x, & \text{if the variable } x_k \text{ is included in the condition } A_j \text{ without the negation} \\ 1, & \text{else} \end{cases}.$$

The function that corresponds to the last numbered variable ($x_{10}$ for $A_2$ in the above example) returns `None` if its argument $x = 0$ thus zeroing out the tensor: a zero value of $x$ means that no member of the condition $A_j$ took the value `True`.

In this problem, the final rank of the constructed tensor depends strongly on the order of the indices, although it should be noted that many SAT algorithms also have heuristics on the sequence of passes over the variables.

## C  RANK OPTIMIZATION

The algorithm presented in the main Theorem 2.1 does not always give optimal ranks.

Below in this section we present additional steps that reduce the ranks. Let's now look more closely at the method of rank reduction briefly described in section 2.3.

In this method, we combine different, but close, values of the derivative functions. Let the values of the derived functions be real and we are given the parameter $\epsilon$ of "smallness". We start by sorting all possible set values of each of the sets $R[i]$, $i = 1, \ldots, d-1$ in Algorithm 1 in ascending order. We then partition all the values into such non-overlapping sets whose difference between a larger and a smaller element does not exceed $\epsilon$. Namely, we put

$$S[i][1] = \{x \in R[i] \colon R[i][1] \le x < R[i][1] + \epsilon\}$$

and then sequentially define

$$S[i][k] = \left\{x \in R[i] \colon \min\left\{R[i] \setminus \bigcup_{j=1}^{k-1} S[i][j]\right\} \le x < \min\left\{R[i] \setminus \bigcup_{j=1}^{k-1} S[i][j]\right\} + \epsilon\right\}$$

until all elements in the initial set $R[i]$ are exhausted. If additionally a maximum rank is given and the number of elements in $S[i]$ exceeds it, we combine the sets further, trying to do so as uniformly as possible.

For each $i$ we update $R[i]$ after defining all sets $S[i]$ at each step:

$$R[i][k] \leftarrow \text{average}\,(S[i][k]), \quad k = 1, \ldots, |S[i]|,$$

where the `average` function can be chosen in different ways, reasonable choices for it are the arithmetic mean and the average between the maximum and minimum elements. Finally, the change we make in the algorithm is to replace the `index_of` function with the following `set_index_of` function

$$z = \text{set\_index\_of}(y, A) \iff y \in A[z].$$

As the second argument, we pass the sets $S$ to this function instead of $R$. Technically, we use the `searchsorted` function from the `numpy` package, and work with interval bounds rather than looking in sets as stated in the definition.

## C.1 DECREASE THE NUMBER OF OUTPUTS

Assume that the middle-index is the last index: $l = d$. This avoids duplicating operations.

If the number of possibles vales of the tensor is small, i. e. the length of the array $R[d]$ is small, we can perform the following trick. Consider the images $(f_{i_d}^{(d)})^{-1}$ of the $d^{\text{th}}$ set of derivative functions

$$(f_{i_d}^{(d)})^{-1}(a) := \{x \colon f_{i_d}^{(d)}(x) = a\}, \quad i_d = 1, \ldots, n_d.$$

The idea of reducing the rank of the TT-decomposition is that the values from this set are indistinguishable for the function $f_{i_d}^{(d)}$. Thus, if there exists a set indistinguishable for all set of the last derivative functions, then its elements can be encoded by a single basis vector. Namely, consider the following array (ordered set) of sets

$$S[d-1] := \left\{ s = \bigcap_{i=1}^{n_d} (f_i^{(d)})^{-1}(a_i) \colon \{a_1, a_2, \ldots, a_{n_d}\} \in (R[d])^{\times n_d}, \ s \neq \varnothing \right\}.$$

The array $S[d-1]$ contains all nonempty sets, that are indistinguishable for functions $f_i^{(d)}$ for any value of $i = 1, \ldots, n_d$. The order in which these sets are included in $S[d-1]$ is unimportant, but it is fixed. The number of all this sets is not greater than the number of outputs in $R[d-1]$: $\text{len}(S[d-1]) \leq \text{len}(R[d-1])$ as each element of $R[d-1]$ belongs exactly to one set in $S[d-1]$. Due to this, a decrease in ranks is achieved.

This procedure is repeated sequentially for each output. At the $k^{\text{th}}$ step we find an array of sets

$$S[k-1] := \left\{ s = \bigcap_{i=1}^{n_k} (f_i^{(k)})^{-1}(A_i) \colon \{A_1, A_2, \ldots, A_{n_k}\} \in (S[k])^{\times n_d}, \ s \neq \varnothing \right\},$$

where $\{A_i\}$ are sets and we define the image on a set as

$$(f_{i_k}^{(k)})^{-1}(A) := \{x \colon f_{i_k}^{(k)}(x) \in A\}.$$

After all sets $S[k]$ ($k = 1, \ldots, d-1$) are found, we construct integer-valued functions by analogy with the functions (5)

$$\hat{f}_i^{(k)}(x) = \text{set\_index\_of}\left(f_i^{(k)}(\hat{x}), \ S[i]\right), \quad \hat{x} \in S[k-1][x], \ q = 1, \ldots, d. \tag{14}$$

We let $S[d] = \{\{x\} \colon x \in R[d]\}$. The definition (14) is correct because the value of the functions $f^{(i)}$ does not depend on the choice of a particular element $\hat{x}$ in the set $S[i-1][x]$.

## C.2 ROUNDING

The cores that are obtained using Algorithm 2 have one important property—they are „almost" orthogonal in the sense that

$$\sum_i \sum_\alpha \mathbf{G}_k(\alpha, i, l) \mathbf{G}_k(\alpha, i, m) = \lambda_l \delta_{lm}, \quad k = 1, \ldots, d-1. \tag{15}$$

This is true, because matrices $Q_k(i) = \mathbf{G}_k(:, i, :)$ consist of rows either identically equal to the zero vector or the basis vector $e(j)^T$ for some $j$. The values of the coefficients $\lambda_l$ have the physical meaning of the number of occurrences of the value $l$ as the value of the function $f^{(k)}$. This number is equal to the number of occurrences of row vector $e(l)^T$ in all matrices $Q_k(i)$. Thus $\lambda_l > 0$. The consequence of relation (15) is the following theorem which is a modified Lemma 3.1 from (Oseledets, 2011):

**Theorem C.1.** *Let $Q_k(i) = \mathbf{G}_k(:, i, :)$, $Q_k(i) \in \mathbb{R}^{r_{k-1} \times r_k}$, $r_0 = 1$, be indexed matrices with tensors $\mathbf{G}_k$ satisfying (15). Let the matrix $\mathbf{Z}$ be defined as follows*

$$\mathbf{Z}(\overline{i_1 i_2 \ldots i_k}, l) := Q_1(i_1) Q_2(i_2) \cdots Q_k(i_k) =$$
$$= \sum_{\alpha_1, \ldots, \alpha_{k-1}} \mathbf{G}_1(1, i_1, \alpha_1) \mathbf{G}_2(\alpha_1, i_2, \alpha_2) \cdots \mathbf{G}_k(\alpha_{k-1}, i_k, l).$$

*Then the matrix $\boldsymbol{Z}$ satisfy the following orthogonality condition*

$$(\boldsymbol{Z}^T \boldsymbol{Z})(l,\, m) = \sum_{\overline{i_1 i_2 \ldots i_k}} Z(\overline{i_1 i_2 \ldots i_k},\, l) Z(\overline{i_1 i_2 \ldots i_k},\, m) = \Lambda_l \delta_{lm}$$

*with natural $\Lambda_l \in \mathbb{N}$.*

Thus, when rounding the tensor with the algorithm described in (Oseledets, 2011), we can start with the second step of this algorithm, skipping the orthogonalization step. In the case of setting the accuracy $\epsilon$ the threshold for discarding the singular numbers must take into account the values $\Lambda_l$.

In the case where we need an exact representation of the given tensor in TT-format, but with the smallest possible ranks, the following consequence helps us.

**Consequence C.1.** The cores, obtained by Algorithm 2, have the optimal ranks if the unfolding of the last core is of full-column rank.

## D  FINDING NON-ZERO ELEMENT IN A TT-TENSOR

Consider the case when the TT-tensor is an indicator tensor of some subset of its index values, i. e. it is equal to one (or an arbitrary values greater than zero) on a small number of combinations of its index values and to zero on the other combinations. The problem is to find at least one combination where the tensor value is greater than zero.

The solution to this problem is based on the algorithmic simplicity of multidimensional summation of the tensor defined in the TT format. Let $\mathsf{I}$ be the tensor of interest. Consider the sum of values of $\mathsf{I}$ on all variables except the first one

$$v_1(i) = \sum_{i_2=1}^{n_2} \cdots \sum_{i_d=1}^{n_d} \mathsf{I}(i,\, i_2,\, \ldots,\, i_d).$$

Knowing $v_1$, we can find out the value of $\hat{\imath}_1$ of the first index of the desired combination:

$$\hat{\imath}_1 = \arg\max_i v_1(i) > 0.$$

Indeed, if $\hat{\imath}_1$ is not a part of any desired combination, then for any values of other indices the value of the tensor is zero: $\mathsf{I}(\hat{\imath}_1,\, i_2,\, \ldots,\, i_d) = 0$. But this contradicts the fact that the value of the sum over these variables is greater than zero.

Then we sequentially find the indices of the desired combination. For the second index:

$$v_2(i) = \sum_{i_3=1}^{n_2} \cdots \sum_{i_d=1}^{n_d} \mathsf{I}(\hat{\imath}_1,\, i,\, i_3,\, \ldots,\, i_d),$$

$$\hat{\imath}_2 = \arg\max_i v_2(i)$$

ans so on. This sequence of steps is summarized in Algorithm 4.

---

**Algorithm 4** Algorithm for calculating indices set of the non-zero value of a TT-tensor

---

**Require:** Cores $\{\mathbf{G}_i\}_{i=1}^d$ of the TT-decomposition of the tensor $\mathsf{I}$ that takes non-negative values
**Ensure:** Set of indices $\{\hat{\imath}_1,\, \hat{\imath}_2,\, \ldots,\, \hat{\imath}_d\}$ s.t. $\mathsf{I}(\hat{\imath}_1,\, \hat{\imath}_2,\, \ldots,\, \hat{\imath}_d) > 0$
 1: **for** $k = 1$ to $d$ **do**
 2:     $\boldsymbol{G}_k \leftarrow \sum_{i=1}^{n_k} \mathbf{G_k}(:,\, i,\, :)$
 3: **end for**
 4: **for** $k = 1$ to $d$ **do**
 5:     $v_k(i) \leftarrow \mathbf{G}_1(1,\, \hat{\imath}_1,\, :)\mathbf{G_2}(:,\, \hat{\imath}_2,\, :)\cdots\mathbf{G_k}(:,\, \hat{\imath}_{k-1},\, :)\mathbf{G_k}(:,\, i,\, :)\boldsymbol{G}_{k+1}\cdots\boldsymbol{G}_d$
 6:     $\hat{\imath}_k \leftarrow \arg\max_i v_k(i)$
 7: **end for**
 8: Return $\{\hat{\imath}_1,\, \hat{\imath}_2,\, \ldots,\, \hat{\imath}_d\}$

---

