# OpenReview forum: "Constructive TT-representation of the tensors given as index interaction functions with applications"
_ICLR.cc/2023/Conference — ICLR 2023 poster_

### Official Review · Reviewer_vyEW · 2022-10-20

**Confidence:** 4
**Correctness:** 3
**Technical Novelty And Significance:** 3
**Empirical Novelty And Significance:** 2
**Recommendation:** 6

**Clarity, Quality, Novelty And Reproducibility:**

The article presents a nice an original method for dealing with high dimensional tensors, although its applicability is limited to a few selected problems.
The quality and clarity of the presentation could be largely improved, some parts are not clear enough and many relevant ideas and results are briefly commented in the paper (see my comments in the Weaknesses section).


**Strength And Weaknesses:**

Strengths
-	The idea is novel, and it could have important applications beyond the scope of the present paper.

Weaknesses
-	The assumption that a set of analytical derivative functions is available is a very strong hypothesis so the number of cases where this method can be applied seems limited.
-	The high dimensional tensor can be also compactly represented by the set of derivative functions avoiding the curse of dimensionality, so it is not clear what is the advantage of replacing the original compact representation by the TT representation. Maybe the reason is that in TT-format many operations can be implemented more efficiently. The paper gives not a clear explanation about the necessity of the TT representation in this case.
-	It is not clear in which cases the minimum rank is achieved by the proposed method. Is there a way to check it?
-	In the paper it is mentioned that the obtained core tensors can be rounded to smaller ranks with a given accuracy by clustering the values of the domain sets or imposing some error decision epsilon if the values are not discrete. It is not clear what is, in theory, the effect on the approximation in the full tensor error. Is there any error bound in terms of epsilon?
-	The last two bullets in the list of main contributions and advantages of the proposed approach are not clear to me (Page 2).
-	The method is introduced by an application example using the P_step function (section 2.2). I found this example difficult to follow and maybe not relevant from the point of view of an application. I think, a better option would be to use some problem easier to understand, for example, one application to game theory as it is done later in the paper.
-	Very relevant ideas and results are not included in the main paper and referred instead to the Appendix, which makes the paper not well self-contained.
-	The obtained performance in terms of complexity for the calculation of the permanent of a matrix is not better than standard algorithms as commented by the authors (Hamilton walks obtained the result with half of the complexity). It is not clear what is the advantage of the proposed new method for this application.
-	The comparison with the TT-cross method is not clear enough. What is the number of samples taken in the TT-cross method? What is the effect to increase the number of samples in the TT-cross method. I wonder if the accuracy of the TT-cross method can be improved by sampling more entries of the tensor.

Minor issues:
-	Page 2: “an unified approach” -> “a unified approach”
-	Page 2: “and in several examples is Appendix” -> “and in several examples in the Appendix”
-	In page 3, “basic vector e” is not defined. I think the authors refers to different elements of the canonical base, i.e., vectors containing all zeros except one “1” in a different location. This should be formally introduced somewhere in the paper.
-	Page 9: “as an contraction” -> “as a contraction”


**Summary Of The Paper:**

This paper presents a method to build a Tensor Train (TT) representation of a multidimensional tensor when the computation of the tensor entries has an analytical expression with a tree structure involving derivative functions that apply to single entries and values of their neighbor derivative functions (Fig. 1a). Moreover, the obtained representation is exact and it has low TT-ranks when the domain sets of the derivative functions are discrete and have small cardinality. The advantage of the proposed method is that, once the TT model is available, it is possible to operate with it in an efficient way for example to implement sums or convolutions of high-dimensional tensors directly in the TT format. The authors apply their method to several combinatorial problems, including game theory problems and to the calculation of the permanent of a matrix.

**Summary Of The Review:**

I think that, although the paper could be highly improved in quality and clarity, the ideas and method introduced are interesting and could have large impact in future applications.

---

> ### Author Response · Authors · 2022-11-18
> **Minor issues and typos**
>
> We thank the reviewer for these inaccuracies. We corrected these typos in our text and gave a definition of the basis vector in a footnote (in the recommendations for the articles preparation there are given instructions for the designation of frequently occurring quantities. The designation of the basis vector is among them, so we thought that this designation will be known to readers).

---

> > ### Comment · Reviewer_vyEW · 2022-12-09
> > **One more typo**
> >
> > Thank you for considering my suggestions.
> > I have found a new typo in the las version of the paper.
> > In Section C, there is a missing cross reference (?).

---

> ### Author Response · Authors · 2022-11-18
> **The comparison with the TT-cross method is not clear enough.**
>
> We thank the reviewer for his question. We have not counted the number of black-box calls to the cross method in the case of cooperative games. The cross algorithm has a parameter $\epsilon$, which indirectly determines the approximation accuracy and number of train samples. The smaller it is, the more accurate the resulting TT-tensor is, but the more time it takes to build it.
>
> We took the parameter $\epsilon$ ($=10^{-2}$) from Ballester's paper, and it turned out to be somewhat optimal: at this parameter we already obtain good accuracy, and when we reduce it to $\approx 10^{-5}$ the accuracy is almost unchanged.
>
> But we want to emphasize that in our experiments with cooperative games we were comparing running times. Our algorithm yields machine accuracy, and it is significantly faster than the cross-based method. If the accuracy of cross were improved by increasing the training sample, its running time would be significantly longer, and the comparative graph in terms of time would be even larger in our favor. In these examples, we show that our method is both more accurate and faster than the methods in Ballester's paper.

---

> > ### Comment · Reviewer_vyEW · 2022-12-09
> > **Parameter epsilon in Ballester's paper**
> >
> > Thank you for your response.
> > I think that a better comparison analysis would be to considering Ballester's TT-cross methods at different epsilon values because it is possible that using even a worse (larger) epsilon parameter, the accuracy is still good and running times faster.

---

> ### Author Response · Authors · 2022-11-18
> **The obtained performance in terms of complexity for the calculation of the permanent of a matrix is not better than standard algorithms as commented by the authors (Hamilton walks obtained the result with half of the complexity). It is not clear what is the advantage of the proposed new method for this application.**
>
> We thank the reviewer for this question. With the matrix permanent example, we want to show that using our universal approach, one obtains almost the same computational complexity as **ad-hoc** approaches applied to calculating of the permanent. We applied our approach practically head-on, and we apply our algorithm to many other examples in the same way. On the other hand, if one computes matrix permanent head-on, the complexity of this approach is noticeably worse than ours. And only if one uses special procedures for calculating permanent, such as Hamiltonian walks, then the complexity of this approach and ours are approximately equal (the number of operations differs by a factor of only 2).

---

> > ### Comment · Reviewer_vyEW · 2022-12-09
> > **Permanent of a matrix computation complexity**
> >
> > Thank you for your response.
> > However, I still found this example not very relevant since the achieved complexity is not better than available alternative methods.

---

> ### Author Response · Authors · 2022-11-18
> **Very relevant ideas and results are not included in the main paper and referred instead to the Appendix, which makes the paper not well self-contained.**
>
> We agree. In the main text of the paper we emphasized the theoretical basis of our method, and gave only the most, from our point of view, revealing examples. Most of the examples ended up in Appendix due to the limited space of the paper.

---

> > ### Comment · Reviewer_vyEW · 2022-12-09
> > **Not well-self contained**
> >
> > Thank you for the explanation. I understand that space limitation makes the writing challenging. Unfortunately the paper is not easy to follow without reading the appendix.

---

> ### Author Response · Authors · 2022-11-18
> **The method is introduced by an application example using the $P_{\hbox{step}}$ function (section 2.2). I found this example difficult to follow and maybe not relevant from the point of view of an application.**
>
> Thank you for pointing that out. Indeed, the step function example is quite non-trivial. But with this example we wanted to show, among other things, that even when at first sight it seems that the dependence of the tensor value on its indices does not decompose into a sequence of derivative functions, such a sequence can exist. And what is more, in this example this sequence leads to a small TT-rank of 2. Very simple examples are given in Appendix (e.g. sum of values), they are not included in the main text due to space limitations.

---

> > ### Comment · Reviewer_vyEW · 2022-12-09
> > **Step function example**
> >
> > Thank yo for your response. I understand the authors motivation for introducing this example. However, I still think that it makes the paper difficult to read and a better option would be to use a more easy to understand example such as one of the game theory used later in the paper.

---

> ### Author Response · Authors · 2022-11-18
> **The last two bullets in the list of main contributions and advantages of the proposed approach are not clear to me (Page 2).**
>
> Thank you for pointing this out.
> We remove the last bullet,
> and the penultimate bullet has been rewritten to read as follows
>  ``the possibility in some cases to explicitly reduce the set of matrix operations on the TT-cores in calculating the required value to an iterative algorithm, since the cores of TT-decomposition are sparse and their elements can be constructed explicitly in advance.``

---

> > ### Comment · Reviewer_vyEW · 2022-12-09
> > **Clarity on advantages summary is improved**
> >
> > Thank you for considering my comments on the clarity of the provided summary of advantages and modifying the text in the revised version.

---

> ### Author Response · Authors · 2022-11-18
> **...the obtained core tensors can be rounded to smaller ranks with a given accuracy by clustering the values of the domain sets or imposing some error decision epsilon...Is there any error bound in terms of epsilon?**
>
> Thank you for this question. The connection between the error of the resulting tensor and the epsilon parameter has turned out to be rather complicated and not obvious as our experiments suggest. This research will be the subject of future work. We decided not to include the preliminary results of the rounding version of Algorithm in this paper.

---

> > ### Comment · Reviewer_vyEW · 2022-12-09
> > **rounding version of the algorithm not supported**
> >
> > Thank you for your feedback.
> > I think the idea of clustering the values of the domain sets through an error decision epsilon is not well supported by theory or experiments so I agree with the authors decision of not including that in the paper. It is well known that, even though a small epsilon is considered, for high dimensional tensors those errors can be greatly amplified resulting in a non satisfactory tensor approximation error bound.

---

> ### Author Response · Authors · 2022-11-18
> **It is not clear in which cases the minimum rank is achieved by the proposed method. Is there a way to check it?**
>
> We agree that we cannot say in advance whether the rank obtained will be optimal. The rank depends on how the derivative functions were chosen, and they may not be chosen in the only way, as shown, for example, in the examples with cooperative games. Optimising our approach to, among other things, reduce the resulting rank is the subject of future work.
>
> When the TT-tensor is already constructed, we can check its ranks for optimality using the procedure `truncate`, described in (Oseledets, 2011) and implemented, for example, in `ttpy` package. If after its work (with zero epsilon) the ranks do not change, it means that the original ranks were optimal.

---

> > ### Comment · Reviewer_vyEW · 2022-12-09
> > **minimum rank check**
> >
> > I thank the authors for providing an explanation on how to check in practice if TT-ranks cannot be further reduced.

---

> ### Author Response · Authors · 2022-11-18
> **The high dimensional tensor can be also compactly represented by the set of derivative functions avoiding the curse of dimensionality...Maybe the reason is that in TT-format many operations can be implemented more efficiently. The paper gives not a clear explanation about the necessity of the TT representation in this case.**
>
> Thank you for pointing this out. The main advantage of the TT decomposition used in our work is the realised tensor algebra over tensors in TT format. The Introduction states:
> ``The advantage of representing tensors in the TT format is not only in overcoming the curse of dimensionality, but also in the implemented tensor algebra for them: we can easily add, multiply, round TT-tensors, calculate the convolution (Oseledets, 2011). In this way we can, for example, construct a set of indicator tensors that represent some constraints in the given problem in advance, and then combine these constraints arbitrarily by multiplying these tensors with a data tensor. As a practical use of such a scheme, we give an example of calculating the permanent of a matrix.''
>
> In the examples dealing with cooperative games we use tensor convolution (the sum of all its elements), which has an efficient algorithm for tensors in TT format. We added the corresponding text to the new version of the paper to the Introduction.

---

> > ### Comment · Reviewer_vyEW · 2022-12-09
> > **TT-format as a convenient operational representation tool**
> >
> > I thank the authors for confirming my understanding about the convenience of representing tensors with the TT format even is a hierarchical representation given by the set of derivative functions is available.

---

> ### Author Response · Authors · 2022-11-18
> **The assumption that a set of analytical derivative functions is available is a very strong hypothesis so the number of cases where this method can be applied seems limited.**
>
> Thank you for pointing this out. Indeed, our method is only applicable if we manage to construct a sequence of derivative functions. Moreover, if we want to have a small rank, we additionally impose a condition on the smallness of the images of these functions. However, as the examples in the main text of the paper and in Appendix show, the said sequences can be constructed for quite a large number of practically interesting problems. We conjecture that in the vast majority of cases, when the TT-decomposition exists at all, we can find an appropriate sequence of derivative functions, but a rigorous justification of this fact is a point of future work.

---

> > ### Comment · Reviewer_vyEW · 2022-12-09
> > **Rigorous justification is missing**
> >
> > I thank the authors for providing a response to this issue.
> > I agree that it seems that it is possible to apply this approach to a vast majority of cases but a rigorous analysis should be performed in a future work.

---

### Official Review · Reviewer_ApYB · 2022-10-21

**Confidence:** 3
**Correctness:** 3
**Technical Novelty And Significance:** 3
**Empirical Novelty And Significance:** 3
**Recommendation:** 6

**Clarity, Quality, Novelty And Reproducibility:**

**Clarity and quality**

The clarity and quality of this paper are relatively lower than others I review or read from the same-level conferences.

1. some technical vocabulary used in the paper is not well explained. For example, the definition of “analytic dependence” is not clearly given throughout the paper, but it seems to be a crucial concept to understand the main idea.
2. The proposed method is only suitable for tensors, of which the image of the involved “derivative function” is small-size (proved in Thm 2.1), but there is no discussion about which families of functions result in such property. The lacking of discussion on this point makes me doubt if the proposed method can be widely applied in real.
3. I agree that the applications mentioned in the paper are very interesting and would inspire lots of tensor researchers. However, in the experiment of the cooperative games, only the work~(Ballester-Ripoll, 2022) is implemented for comparison. It is difficult to support that the proposed method outperforms SOTAs claimed in conclusion.

**Novelty**:

The novelty of this paper seems good. The paper focuses on a relatively different setting of the tensor network, unlike tensor decomposition, completion, and parameter compression, which has been widely discussed in the machine learning community.

In my own opinion, this work is closely similar to studies by Boris N. Khoromskij et al. on quantized tensor approximation (whose paper was also cited by the authors). For example in sec 4 of (Khoromskij, 2018), it theoretically discussed the low TT-rank property of various functions under reshaping. I do not find too much difference between this paper and these existing works. It will be good if the authors highlight the differences in the revision of the manuscript.

*Khoromskij, Boris N. "Tensor numerical methods in scientific computing." Tensor Numerical Methods in Scientific Computing. De Gruyter, 2018.*

**Reproducibility**

The experimental codes were provided.

**Strength And Weaknesses:**

+Strength:

1. The targeted problem is relatively novel. In machine learning, the problem is rarely discussed in the tensor community.
2. The applications mentioned in the paper—cooperative games and the computation of matrix permanent—illustrate the potential of tensor methods in new applications.

- Weakness

1. The paper is hard to follow, even though the main idea seems not complicated.
2. The experimental results for the cooperative game are not convincing.

**Summary Of The Paper:**

This paper studied a new way to build the tensor train (TT) representation for a special group of functions. In particular, as proved in the paper, this paper claimed that a functional tensor could be represented in the low-rank TT format if the image of the “derivative functions” is sufficiently small. In the experiments, the proposed method was used for two applications: (1) cooperative games and (2) the computation of the matrix permanent. The proposed method gives a significant improvement in the performance for both two problems.

**Summary Of The Review:**

It is hard to simply judge if this work is good or bad. It’s quite different from others I have reviewed recently. Both the strength and drawbacks are obvious. It would be a good work if the mentioned problem is revised.

---

> ### Author Response · Authors · 2022-11-18
> **this work is closely similar to studies by Boris N. Khoromskij et al. ... It will be good if the authors highlight the differences in the revision of the manuscript.**
>
> Thank you for pointing this out. We have changed the text in ''Related work'' to the following:
> ``In the paper (Oseledets, 2012) explicit representations of several tensors with known analytical dependence of indices are presented, bur for a fairly narrow class of tensors (see also (Khoromskij, 2018, Sec. 4.2)).``
>
> The main difference of our work is that we can construct a TT-decomposition for any dependence of the tensor value on its indices in case we can construct a suitable sequence of derivative functions. In the cited works only a limited set of functions for which there is an explicit representation in TT format is considered, without the possibility of a global generalization. Our numerous examples show that it is possible to construct a "good" sequence of derivative functions for a large number of problems.

---

> > ### Comment · Reviewer_ApYB · 2022-12-03
> > **Thanks for the response.**
> >
> > I'm convinced that the (promised) revision of the manuscript would make it more clear and easier to follow.
> >
> > Best,

---

> ### Author Response · Authors · 2022-11-18
> **It is difficult to support that the proposed method outperforms SOTAs claimed in conclusion. However, in the experiment of the cooperative games, only the work~(Ballester-Ripoll, 2022) is implemented for comparison.**
>
> We thank the reviewer for his comment. Indeed, as SOTA we were referring to the above article by Ballester-Ripoll, which is quite recent. We would be happy to learn about other approaches for the cooperative games, that work better and compare, but we are not aware about other (maybe non-tensor?) baselines.

---

> ### Author Response · Authors · 2022-11-18
> **The proposed method is only suitable for tensors, of which the image of the involved “derivative function” is small-size...  but there is no discussion about which families of functions result in such property.**
>
> We thanks the reviewer for this comment. We agree that there is as no rigorous theory yet which can show in which cases it is possible to construct a sequence of functions in which the range of values of the derivative functions would be small so that the resulting TT-decomposition would be low-rank. However, there is every reason to believe that this is possible in the vast majority of cases where a low-rank decomposition of a given tensor (which can be obtained by other methods) exists in principle.
>
> Moreover, as the example with the computation of the matrix permanent shows, even in the case where the ranks of the TT-decomposition are formally quite large, our approach makes sense. Indeed, in the permanent example, our approach reduces the computation of the given value to an iterative algorithm based on an explicit representation of the cores of the TT-decomposition, and due to the sparse structure of these cores, the algorithm is still effective despite the initial large ranks.
>
> To be honest, we cannot say for ourselves how broadly the approach outlined in the article is applied. We sent the publication to the theoretical session of the conference. We believe that a detailed study of possible applications and the practical application of our method is the subject of future work.

---

> ### Author Response · Authors · 2022-11-18
> **the definition of “analytic dependence” is not clearly given**
>
> Thank you for pointing this out. We add the following footnote to the text:
> ``By analytic dependence we mean the known symbolic formula for the tensor value, not the definition of the term within complex analysis.''

---

> ### Author Response · Authors · 2022-11-18
> **The paper is hard to follow**
>
> We thank the reviewer for the comment. We have made a few changes to the structure of the article to make it more readable. We have moved Section 3.3 to the Introduction. In this way, we immediately list the tasks where our approach has been or can be applied. We believe that this changes will help the reader to understand the main purpose of our research. In addition, for the same reason, we have put the figure with example of 10-queen problem closer to the Introduction and describe it in the Introduction.

---

### Official Review · Reviewer_EMLj · 2022-10-22

**Confidence:** 3
**Correctness:** 4
**Technical Novelty And Significance:** 3
**Empirical Novelty And Significance:** 3
**Recommendation:** 6

**Clarity, Quality, Novelty And Reproducibility:**

I struggled to understand what they are exactly referring to "their method". I believe they want Algorithm 2 to be "their method" as that is the algorithm that takes the left and right set of integer-valued derivative functions to TT-cores.

While there are lots of papers that work with tensor formats of functions and develop techniques for compression, they most work in the lossy setting. The paper is demonstrating the approach in setting for which TT formats are rarely employed.

**Strength And Weaknesses:**

The main observation in this paper is in Figure 1: A computational tree for the values of a tensor gives you a TT-decomposition. Going in the opposite direction is standard. A TT-decomposition gives you an efficient computational tree to recover any tensor entries. However, the usual challenge is finding a reasonable computational tree. In this paper, the authors give many applications where one can write a reasonably efficient computational tree. Before reading this paper, I always imagined that writing down reasonable computational trees for each entry of a tensor was challenging except in very simple examples. That's why we have to compute the TT format.

However, the authors demonstrate this semi-analytic technique on several applications that ends up being quite convincing. In particular, the appendix contains many applications and examples of derived derivative functions. It would be nice to have more discussion on how one can construct a reasonable derivative functions (hence, computational tree). Or, are the authors just writing down anything that works? When one can write down a reasonable set of left and right derivative functions, then the resulting TT-decomposition is usually sparse.

The big weakness of the paper is the possibility that it is quite a specialized approach that only works on a handful of carefully selected functions, f, for the tensor entries.  It is hard to judge the generality of the approach, but the range of applications in the appendix is highly welcomed. I also assume that it is hard to write down a computational tree, in the setting when you are happy with a lossy compression.

In some of the applications in the appendix, it was unclear what the computational benefit is in terms of solving a problem.

**Summary Of The Paper:**

Given a function f : Z^d -> R, the authors construct tensor-train representations for the implicitly defined tensor. They first need to build a computational tree for the entries of the tensor that involves left and right derivative functions. From this, they show that they can construct a TT format for the tensor X_{i1,...,id} = f(i1,...,id) that has a TT-rank vector that depends on the size of the image of each derivative function (see Theorem 2.1 and Algorithm 2). They use analytical techniques to write down the derivative functions. Once the tensor is in TT-format, subsequent tensor computations can be sped up.


**Summary Of The Review:**

The paper presents a relatively simple, but effective, idea of deriving TT decompositions of tensors by analytically writing down a computational tree. The scope of this paper may be slightly limited, though in the appendix the authors do a good job at demonstrating the approach on a range of problems.

---

> ### Author Response · Authors · 2022-11-18
> **... what they are exactly referring to "their method".**
>
> We thank the reviewer for this remark. By ``our method'' we meant the whole method of constructing a TT decomposition according to a given dependence of a tensor value on its indices. However, when considering specific applications, we may have meant by this term a broader concept: the construction and specific mechanism for using a TT decomposition to calculate a given value.  We agree that a double reading is possible here, but we believe that it is clear from the context what we are talking about in each case.

---

> ### Author Response · Authors · 2022-11-18
> **In some of the applications in the appendix, it was unclear what the computational benefit is in terms of solving a problem.**
>
> Thank you for pointing this out. We agree that many of our examples do not demonstrate computational superiority. They have been added to show that our universal approach to constructing a TT-decomposition, together with the application of such a decomposition to calculating any given quantity, is quite universal. We think that our main contribution is theoretical, but we have demonstrated that it works for interesting real-life problems.

---

> ### Author Response · Authors · 2022-11-18
> **I also assume that it is hard to write down a computational tree, in the setting when you are happy with a lossy compression**
>
> We thank the reviewer for this comment. In the lossy compression case, the computation tree remains the same as well as the sequence of derivative functions, but the procedure of constructing the cores of the TT-decomposition changes. However, we agree that investigating the lossy compression case is the subject of further research, and we have little information about such a case now.

---

> ### Author Response · Authors · 2022-11-18
> **...it is quite a specialized approach that only works on a handful of carefully selected functions, f, for the tensor entries.**
>
> Thank you for your comment. You have raised an important point here. However, we believe that our approach allows us to construct a TT-decomposition for a large number of functions. This is confirmed by numerous examples in the main text of the paper and Appendix. The main difficulty lies in constructing a sequence of functions that would be obtained from a given analytic dependence of the tensor value as a function of its indices.
>
> We agree that there is as no rigorous theory yet which can show in which cases it is possible to construct a sequence of functions in which the range of values of the derivative functions would be small so that the resulting TT-decomposition would be low-rank. However, there is every reason to believe that this is possible in the vast majority of cases where a low-rank decomposition of a given tensor (which can be obtained by other methods) exists in principle.
>
> Moreover, as the example with the computation of the matrix permanent shows, even in the case where the ranks of the TT-decomposition are formally quite large, our approach makes sense. Indeed, in the permanent example, our approach reduces the computation of the given value to an iterative algorithm based on an explicit representation of the cores of the TT-decomposition, and due to the sparse structure of these cores, the algorithm is still effective despite the initial large ranks.

---

### Official Review · Reviewer_gzvu · 2022-10-27

**Confidence:** 4
**Correctness:** 4
**Technical Novelty And Significance:** 2
**Empirical Novelty And Significance:** 2
**Recommendation:** 6

**Clarity, Quality, Novelty And Reproducibility:**

Clarity
The paper can be followed by someone who has a strong tensor-train background, but the existing presentation would be difficult for someone without it. The paper might benefit from a slower description and motivation in section 2 and 2.1, where it is somewhat unclear why the "middle-out" structure is needed, especially if "it is more efficient to start the calculation from one end" (bottom of page 2).

Quality
With the appendix, the construction and exploration of different problems is significant.

Novelty
The formulation of various games as TT representation seems to be novel and interesting. However, the practical value of the proposed method seems quite low, outside of curiosity and interest.

Reproducibility
I feel confident that, with a careful reading, anyone would be able to fully reproduce the work. The provided code is sufficient to replicate all experiments with relative ease.

**Strength And Weaknesses:**


Strengths
1) The paper presents an interesting formulation building on existing work modeling functions as tensor representations.
2) A large number of (cool!) example problems are provided, which are both individually interesting and strongly support the author's claims that the method is quite general and can be variously applied.
3) All formulations and implementations are extremely detailed, and an interested reader would have no problem recreating this work given the paper and Appendix, as well as the anonymized linked code.

Weaknesses
1) The organization of the paper could be improved. In the main paper, the Appendix is referenced 11 times. Many of those places seem to allude to significant contributions or problems of interest. Personally, I was able to much more understand the motivation and the construction of the method through some of the examples given in the Appendix (knapsack, n-queens). Throughout the main paper as a reader, it was hard to understand and follow the motivation of why and where this could be valuable or interesting. The cooperative game examples were a bit further outside of my familiarity, and I assume this might be true for other readers; including a classical combinatorial optimization or other Hard problem from the Appendix could connect with a wider audience. Sections 3.3 and 4 could be merged with the introduction, they do not meaningfully add much currently.

2) Theorem 2.1 does not seem to be revealing anything particularly interesting, a discussion of why this is not obvious or why it is valuable would be beneficial over the full proof which could be moved to the Appendix, leaving more space for additional examples.

3) In the end, the practical value is not demonstrated that well. While the construction performs better than the cross-TT approach for the given problems, it's not clear why constructing functions in this way is particularly valuable over the existing, "non-tensor" problem formulations or functions. The complexity of the tensorized problem in both space and time seems to only be lower bounded by existing solutions/solvers. This is further demonstrated by the paper's own description that the rank can sometimes be low and sometimes significantly higher than approximation methods. The experiments do not compare solving the problem against traditional solvers.


**Summary Of The Paper:**

The paper presents a method for formulating particular multivariate functions based on indices as a particular tensor train decomposition. An overview of the construction is presented, where so-called derivative functions are used to define index dependences as well as the cores of the TT representation. Analysis of the construction is presented, and a number of experimental settings are evaluated where particular problems can be framed within the proposed derivative function TT framework.

**Summary Of The Review:**

Ultimately the paper is poorly organized, and fails to effectively motivate a reader as to why they should be interested in and value the proposed method. There are some particularly interesting pieces, but they are not communicated effectively in the current presentation.

---
After Rebuttal:
The authors have addressed a majority of my concerns (and mistaken confusions) and put significant effort in restructuring the paper for clarity and readability. It is still difficult for me to see the significance (theoretically or practically) outside of what is presented here, but it is interesting work nonetheless. The revision is much clearer to me;  I have increased my score to reflect this.

---

> ### Author Response · Authors · 2022-11-18
> **...it is somewhat unclear why the "middle-out" structure is needed, especially if "it is more efficient to start the calculation from one end" (bottom of page 2).**
>
> We thank for pointing this out. In the computational tree diagram in Figure 1, as in formulae (1)--(3), the computation takes place starting from the ends. The "middle"-function $f^{(l)}$ is calculated at the last moment when the values of all the other functions in the sequence are known. Such a computational scheme corresponds to the fast computation of the value of an element of the TT-tensor, and thus inherits the properties of the latter.

---

> > ### Comment · Reviewer_gzvu · 2022-12-08
> > **Acknowledged.**
> >
> > I've gone back and agree with the authors here, I believe I misunderstood this piece.

---

> ### Author Response · Authors · 2022-11-18
> **In the end, the practical value is not demonstrated that well... This is further demonstrated by the paper's own description that the rank can sometimes be low and sometimes significantly higher than approximation methods.**
>
> We thank the reviewer for this comment. We think that main contribution is theoretical, but we have demonstrated that it works for interesting real-life problems.
> For the same reasons, we placed our work in the theoretical section of the conference. The motivation for the research was the increasing use of tensor trains, including in applications related to deep learning, which we write about in the introduction.

---

> ### Author Response · Authors · 2022-11-18
> **Theorem 2.1 does not seem to be revealing anything particularly interesting...**
>
> We respectfully disagree that the proof of the Theorem 2.1 is not of practical interest. A detailed description of the algorithm can be found in Appendix in Algorithm 1--2. But the proof of Theorem 2.1 contains the essence of our method. We decided to leave it in the main text, since we want to focus on the theoretical part of our research rather than on its practical application, which is more the subject of future papers.

---

> > ### Comment · Reviewer_gzvu · 2022-12-08
> > **Acknowledged.**
> >
> > I agree that Theorem 2.1 is the core contribution and necessary for the method and the paper, and is more important and nontrivial than I had previously thought. I would push back slightly on the theoretical focus, given that "with applications" is in the paper title and most of the motivation seems to be from practical applications.

---

> ### Author Response · Authors · 2022-11-18
> **The organization of the paper could be improved. ... I was able to much more understand the motivation and the construction of the method ... The cooperative game examples were a bit further outside of my familiarity, and I assume this might be true for other readers ...  Sections 3.3 and 4 could be merged with the introduction, they do not meaningfully add much currently.**
>
> We thank the reviewer for the comment. We have moved Section 3.3 to the text of the Introduction.
> In this way, we immediately list the tasks where our approach has been or can be applied, which we believe will motivate the reader to read further. In addition, for the same reason, we have moved the figure with example of 10-queen problem closer to the Introduction and describe it in the Introduction.
> Also we put Section 4 ``Related work'' to the Introduction.
>
> We also decided to keep examples with cooperative games in the main part of the article because they show the advantage of our approach in both time and speed compared to the cross method from Ballester's recent article. Many other examples in Appendix contain mostly examples of applying our universal approach to various types of problems, but, unfortunately, without demonstrating strong computational advantage.

---

> > ### Comment · Reviewer_gzvu · 2022-12-08
> > **Appreciate the restructuring effort!**
> >
> > I appreciate your effort in restructuring based on mine and others' comments.

---

### Decision · Program_Chairs · 2023-01-20

**Decision:**

Accept: poster

**Justification For Why Not Higher Score:**

The practicality of the approach might attract less attention in this conference

**Justification For Why Not Lower Score:**

The approach is novel and interesting by itself. The authors, through discussions, managed to flip some initially negative reviews, leading to an overall positive set of reviews.

**Metareview: Summary, Strengths And Weaknesses:**

Summary:

The paper presents a method for formulating particular multivariate functions based on indices as a particular tensor train decomposition. An overview of the construction is presented, where so-called derivative functions are used to define index dependences as well as the cores of the TT representation. Analysis of the construction is presented, and a number of experimental settings are evaluated where particular problems can be framed within the proposed derivative function TT framework.

Strengths:

1. The paper presents an interesting formulation building on existing work modeling functions as tensor representations.
2. Detailed explanations, which are both individually interesting.
3. Novelty of approach.

Weaknesses (initially):

1. Presentation of the paper
2. Practicality of the proposed algorithm / narrow set of applications.

Recommendation:

This is a paper that lies on the positive side of the review process. It is quite positive that authors have successfully addressed most (if not all) of the questions, suggesting changes in the final paper. We rely on the authors' discretion include all the proposed changes and improve the quality of the paper in its final version.

**Note From Pc:**

if the above contains the word "oral" or "spotlight" please see: "oral" presentation means -> notable-top-5% and "spotlight" means -> notable-top-25%. As stated in our emails, we are disassociating presentation type from AC recommendations